# Green Agricultural Products Supply Chain Subsidy Scheme with Green Traceability and Data-Driven Marketing of the Platform

**DOI:** 10.3390/ijerph20043056

**Published:** 2023-02-09

**Authors:** Xue Wang, Jiayuan Zhang, Deqing Ma, Hao Sun

**Affiliations:** School of Business, Qingdao University, Qingdao 266071, China

**Keywords:** green agricultural products supply chain, green subsidy, platform traceability, data-driven marketing, differential game

## Abstract

Government subsidies have played an important role in the development of green agriculture. In addition, the Internet platform is becoming a new channel to realize green traceability and promote the sale of agricultural products. In this context, we consider a two-level green agricultural products supply chain (GAPSC) consisting of one supplier and one Internet platform. The supplier makes green R&D investments to produce green agricultural products along with conventional agricultural products, and the platform implements green traceability and data-driven marketing. The differential game models are established under four government subsidy scenarios: no subsidy (NS), consumer subsidy (CS), supplier subsidy (SS), and supplier subsidy with green traceability cost-sharing (TSS). Then, the optimal feedback strategies under each subsidy scenario are derived using Bellman’s continuous dynamic programming theory. The comparative static analyses of key parameters are given, and the comparisons among different subsidy scenarios are conducted. Numerical examples are employed to obtain more management insights. The results show that the CS strategy is effective only if the competition intensity between two types of products is below a certain threshold. Compared to the NS scenario, the SS strategy can always improve the supplier’s green R&D level, the greenness level, market demand for green agricultural products, and the system’s utility. The TSS strategy can build on the SS strategy to further enhance the green traceability level of the platform and the greenness level and demand for green agricultural products due to the advantage of the cost-sharing mechanism. Accordingly, a win-win situation for both parties can be realized under the TSS strategy. However, the positive effect of the cost-sharing mechanism will be weakened as the supplier subsidy increases. Moreover, compared to three other scenarios, the increase in the environmental concern of the platform has a more significant negative impact on the TSS strategy.

## 1. Introduction

Agriculture greenization has been well recognized as an important measure to promote the sustainable development of agriculture. A typical example is that organic agriculture has been initiated and developed rapidly around the world due to environmental and health concerns in the last decades. Moreover, due to the fact that green agricultural products have the characteristics of high safety and rich nutrition, consumers’ purchase intentions for green agricultural products have significantly increased in recent years. According to the *Report on the Current Situation of Public Green Consumption in China (2019 Edition)*, 83.34% of the respondents are willing to buy green agricultural products even at a higher price. Thus, both green sustainability concerns and market consumption trends have become important driving forces of green innovation for agricultural products. However, in practice, some agricultural product suppliers are still reluctant to switch to green technology production [1]. The reason could be that green production requires more labor and costs compared to conventional production [2]. Thus, although consumers’ willingness to pay more for green agricultural products can bring a higher premium to the suppliers than conventional agricultural products, whether the price premium is enough to make up for the increased investments in green technology development remains unclear [3,4,5,6].

In the face of the above problems, governments around the world have developed various policies to ensure the sustainable development of agriculture, among which subsidies have become an essential incentive scheme [7]. As early as 1985, for the healthy and orderly development of agriculture, the United States and Europe implemented the related projects of green agricultural subsidies and formulated agricultural production subsidies such as the “Environmental Quality Incentive Plan”. In 2016, China adopted a reform plan to establish a subsidy system for green ecological agriculture, pointing out that it is necessary to guide the transition from conventional agriculture to organic agriculture through subsidies. At present, the most ordinary subsidy forms include providing subsidies to suppliers and consumers. For example, to mitigate the financial pressure on suppliers of green agricultural products, the Indian government provides subsidies for their investments in green technologies, infrastructure, and exports [8]. The Implementation Plan for Promoting Green Consumption, enacted by the National Development and Reform Commission of China in 2022, suggests that localities should encourage more consumers to participate in green consumption by offering green consumption vouchers, green points, and direct subsidies. Motivated by the above examples, these two subsidy forms are also the focus of this paper.

With the development of information technology, the Internet platform is becoming a new channel to promote the sales of agricultural products, such as on Shunong.com and dbc61.com in China. The platform uses the Internet of Things, big data technology, and visualization technology to share data and realize the traceability of agricultural products. It can not only scientifically and systematically manage the complex information of agricultural products but also provide quality control of the whole process of agricultural products from farmland to table to ensure the safety of products [9,10]. For example, “Ming Jing”, a traceability platform for agricultural products, is based on cloud computing and cloud service technology and uses the characteristics of blockchain, such as multi-participation and non-tampering, to build a data link that runs through all processes of agricultural production, processing, and consumption. Through the combination of the Internet of Things and blockchain, it provides source reliable data support for breeding industry, production enterprises, and consumers (https://blog.csdn.net/hongzaokeji/article/details/123864240 (accessed on 5 February 2023)). Therefore, the traceability of the platform can benefit the agricultural product suppliers, the consumers, and the whole supply chain [11]. In addition, the platform has sufficient ability to collect data, based on which it can predict, analyze, and guide consumers’ purchasing behaviors [12]. Yunnan Baiyao cooperates with the Alibaba platform for data analysis, makes use of big data and the star effect for online marketing, and quickly improves its brand awareness. Additionally, Uniqlo realizes “zero inventory” through platform data analysis (https://wenku.baidu.com/view/437bfcee4328915f804d2b160b4e767f5acf803f.html?_wkts_=1675497004565&bdQuery=%E5%B9%B3%E5%8F%B0%E6%95%B0%E6%8D%AE%E8%90%A5%E9%94%80%E7%9A%84%E6%A1%88%E4%BE%8B (accessed on 5 February 2023)). Such marketing activities that utilize the platform data analysis capability are called platform data-driven marketing (DDM) activities. These data-driven marketing (DDM) activities of the platform can also promote green products and improve the market share of green products [13]. However, since product traceability and DDM activities need extra investments, the platform should determine the appropriate traceability and DDM levels. 

Based on the above observation and analysis, this paper examines the operational decisions for a two-level green agricultural products supply chain (GAPSC) system under different subsidy schemes from a dynamic perspective. Specifically, the supplier adopts a hybrid production mode, i.e., produces conventional and green agriculture products simultaneously. The platform provides green traceability services only for green agricultural products, and DDM services for both types of agricultural products. Four types of subsidy schemes are discussed: (1) no government subsidy; (2) the government only subsidizes the supplier for its green innovation; (3) the government only subsidizes the consumers for their purchase of green products; and (4) the government subsidizes the supplier for its green innovation and the supplier also shares a portion of the platform’s green traceability investment. We focus on the following research questions: (i).Under different subsidy scenarios, how do key parameters, such as consumer green awareness, substitutability between conventional agricultural products and green agricultural products, as well as the relative environment concern coefficients, affect the greenness of agricultural products and the system’s performance?(ii).Which government subsidy scheme is more effective for the green development of GAPSC?(iii).Whether the internal cost-sharing mechanism can strengthen the role of government subsidies or not?

To answer the above questions, we establish Stackelberg differential game models between the supplier and the platform for each subsidy scheme and then derive the optimal feedback equilibrium solutions of the supplier’s green technology investment, the platform’s traceability, and DDM levels based on the Bellman continuous dynamic programming theory. Through a comparative static analysis of key parameters and comparisons among different government subsidy schemes, we draw the conclusion that only when the competition intensity between two types of agricultural products is below a certain threshold consumer subsidy can benefit the GAPSC. Otherwise it will damage the utilities of both parties. Compared to consumer subsidy, supplier subsidy can improve the greenness level of green agricultural products and the system utility more significantly. In addition to the supplier subsidy provided by the government, when the supplier also shares a part of the traceability cost with the platform, a higher greenness level and more system utility can be achieved. 

The contributions of the paper are threefold: (1) to the best of our knowledge, this paper is the first to quantitatively model the impacts of government subsidies on the GAPSC with platform traceability and DDM marketing. Thus, we build the connection between the agricultural supply chain literature and the emerging platform economy literature; (2) we establish differential game models to examine the optimal decisions of the GAPSC from a dynamic perspective; and (3) we find that consumer subsidy is not always beneficial to the supplier and the platform. The supplier subsidy with a cost-sharing mechanism can promote the economic and environmental performance of the GAPSC best.

The rest of this paper is organized as follows: Section 2 analyzes the literature related to this paper, and the differences between this paper and the existing literature are pointed out. Section 3 describes the problem, and relevant assumptions are put forward. Section 4 establishes and solves the differential game models under different subsidy schemes. Section 5 provides a comparative static analysis and a comparative analysis of the models. Section 6 verifies the results with numerical examples. Finally, the discussion and conclusions, as well as future research directions, are drawn in Section 7.

## 2. Literature Review

The research is related to three streams of the literature: the green agricultural products supply chain (GAPSC) management, platform tractability and data-driven marketing, and government subsidy for green innovation in a supply context. Then, we will review the related literature and compare this work to theirs to illustrate our main contributions.

### 2.1. Green Agricultural Products Supply Chain Management

Growing consumer demand for healthy agricultural products and requests from all sectors of society for stricter regulations on production in the agri-food industry have gradually led to the transition from the traditional agri-food supply chain towards a green agricultural products supply chain (GAPSC) [14]. GAPSC can ensure the safety of agricultural products, affect the environment with less negative impact, and optimize the allocation of resources by incorporating “green” or “environmental consciousness” into the traditional agricultural products supply chain [15]. Although the agricultural products suppliers have to invest more in green farming techniques in GAPSC, numerous empirical studies have shown that it has a promising future in that more and more consumers are choosing products that meet their safety and ecological concerns [15,16,17]. They believe that the health and environmental benefits of purchasing green agriculture products are more significant than reducing costs by buying conventional agriculture products [18]. Many scholars have also established quantitative models (mathematical optimization or game theory models) to study a series of operational decision problems in GAPSC. Considering that the introduction of green agricultural products will lead to competition with traditional ones, Ozinci et al. [19] and Perlman et al. [20] explore the pricing decisions of GAPSC for two types of agricultural products in the case of a single distribution channel and dual distribution channels, respectively. Liu et al. [21] find that when manufacturers integrate with retailers to form a centralized GAPSC, the supply chain will achieve the dual goals of environmental protection and profitability. For the phenomenon that many GAPSCs suffer from an inequitable distribution of costs/benefits due to the uneven greening inputs from all parties, Liu et al. [22] construct a model in the context of China’s current agricultural development and propose an appropriate joint investment scheme via cost-sharing and revenue-sharing contracts for big data and blockchain technology adoption. Luo et al. [23] construct differential game GAPSC models considering temperature-controlled inputs under two decision structures: centralized decision and decentralized decision. Their findings suggest that the combined effects of high consumer preferences for green agricultural products and supply chain members’ collaboration contribute to achieving higher economic and environmental performance for the GAPSC.

### 2.2. Platform Tractability and Data-Driven Marketing

Nowadays, Internet platforms have been playing pivotal roles in agriculture supply chain operations [24]. On the one hand, they can utilize emerging technologies (the Internet of Things, RFID, big data, and blockchain) to share related information on products’ safety, quality, and logistics, and track and monitor products along the whole agricultural products supply chain; on the other hand, they can also provide services such as data acquisition, processing, analyzing, forecasting, and guiding consumers’ behaviors based on data-driven analysis for agriculture products [25], i.e., data-driven marketing (DDM). For instance, with the technical support of Ali Cloud and Ant Chain, Alibaba has realized the full traceability of regional brands such as Pu’an red tea and Nan’shan soil honey at the planting end, production end, and consumption end in China. By scanning the QR code provided by Alibaba on the platform, consumers can briefly see the production, quality control, and transportation aspects of the products. Furthermore, Alibaba also uses digital technology to broaden the sales channels for red tea and soil honey and then promote the agriculture industry.

In terms of platform tractability, Steinberger et al. [26] developed a mobile system for collecting farming data that transmits information to a server through the Internet. Yang et al. [27] developed a management platform for improving traceability credibility based on authentication, production management, and supervision information. Qian et al. [28] describe the design and development of a cloud-based platform for rational pesticide use to guarantee the source safety of traceable vegetables. Salah et al. [29] propose an approach that leverages the Ethereum blockchain and smart contracts efficiently to perform business transactions for soybean tracking and traceability across the agricultural supply chain. Hu et al. [30] combine blockchain and edge computing technology to establish a trust framework for an organic agricultural products supply chain. The results show that the framework can not only improve the system performance but also reduce the operational costs compared to traditional techniques.

Recently, more and more Internet platforms have been implementing marketing activities based on data-driven analysis. With DDM, the platform can accurately grasp consumption trends by analyzing historical data and using data-driven technologies to perform real-time calculations, cross-network platform aggregation, and multi-user behavior analysis to find the marketing points that can most stimulate consumers’ purchasing desires and increase their purchasing utilities [12,13,31]. Liu et al. [13] examine a platform’s best choice for the combinations of sales mode (agency selling or reselling) and DDM activity (with or without DDM). They find that, with an improvement in DDM efficiency, the reselling mode is more favorable to the platform than the agency selling mode. Li et al. [32] and Xia et al. [33] investigate government subsidy policies and financing strategies for a green supply chain with the platform’s DDM activities, respectively. However, they do not touch upon the traceability function of the platform in a GAPSC. 

### 2.3. Government Subsidy for Green Innovation

Government subsidies have widely been recognized as an effective instrument to promote green innovation. Currently, the two mainstream subsidy schemes offered by governments are manufacturer/supplier subsidies and consumer subsidies. For the former, it can encourage the manufacturer/supplier to devote more efforts to implementing green innovation and manufacturing green products. Sheu et al. [34] analyze the impacts of the governmental financial intervention on competitive green supply chains and deduce the optimal decisions of the government and supply chain members. It is proven that the government can stimulate manufacturers’ intention to produce green products through taxation and subsidization. Madani et al. [35] develop a Stackelberg game model in which the government acts as the leader and two competitive green and non-green supply chains are the followers, and then discuss optimal pricing strategies, green degree decisions, and governance tariffs. It has been found that government subsidies for green manufacturers have a greater positive impact on supply chain profits and environmental performance than the tax rate. Nielsen et al. [36] compare chain members’ profits, greening levels, consumer surplus, and environmental improvements in the context of different supply chain power structures (manufacturer-Stackelberg and retailer-Stackelberg), different procurement decisions (single-period and two-period), and different government incentives for manufacturers (subsidies on per-unit products and total investment in R&D). Gao et al. [37] establish specific dual-channel green supply chain models to investigate the influences of the government subsidy to the manufacturer and eco-label policy on the environmental benefits of two-type green products (development-intensive and marginally cost-intensive). 

For the latter, it can improve consumers’ purchasing intentions for green products and boost the demand for these products. Ma et al. [38] analyze the impact of consumer-subsidy on the dual-channel closed-loop supply chain and draws the conclusion that the manufacturer, the retailer, and the consumers can benefit from the consumer-subsidy policy. Wang et al. [39] examine the impact of demand forecast information sharing on a green supply chain with government subsidies for consumers. They find that demand information sharing is beneficial to the manufacturer but unfavorable to the retailer. Cohen et al. [40] investigate the joint production and pricing decisions of the supplier facing market demand uncertainty when the government directly subsidies consumers for their purchase of green products. Li et al. [41] analyze two types of government subsidy policies for consumers (consumption subsidies and replacement subsidies) in a dual-channel supply chain and find that both subsidies are beneficial to the manufacturer but intensify the competition between the retail channel and the e-channel. The replacement subsidy is conducive to environmental protection, while the consumption subsidy is good for social welfare. He et al. [42] derive the manufacturer’s optimal pricing strategy and channel structure selection in a dual-channel closed-loop supply chain with consumer subsidies from the government. The results show that the government can guide the manufacturer to choose the preferred channel structure through appropriate subsidy levels. 

Some scholars conduct comparisons between manufacturer subsidies and consumer subsidies. Fu and Chen [43] study the impact of government subsidies on the two-tier supply chain. The government can subsidize suppliers, manufacturers, and consumers through linear subsidies and fixed subsidies. The results show that providing a fixed subsidy to consumers is the best subsidy policy. Yu et al. [44] consider how the allocation of subsidies between consumers and manufacturers can improve consumer welfare and manufacturer profits when the government provides subsidies to the consumers, the manufacturers, or both. Bian et al. [45] probe into and compare the impacts of two types of government subsidies (consumer subsidy and manufacturer subsidy) on the manufacturer’s sales volume and abatement level decisions with environmentally conscious consumers. The results indicate that consumer subsidies yield a lower abatement level but a higher sales volume than manufacturer subsidies. Li et al. [46] investigate the effects of three government policies (no-subsidy, product subsidy to consumers, and innovation subsidy to green manufacturers) under duopoly competition. The conclusion shows that the product subsidy is superior to the innovation subsidy when the green innovation cost is high enough and can make unit production cost reduce remarkably. Chen et al. [47] examine how two types of government subsidies (per-unit production subsidy and innovation effort subsidy) influence a two-echelon research joint venture (RJV) supply chain. For each subsidy scheme, the government endogenizes the subsidy to maximize social warfare. They find that although both the manufacturer and the retailer can benefit from RJV, the government suffers a loss in certain circumstances. Li and Liu [48] investigate the impact of the government subsidy program on the innovation level of the secondary supply chain, and the results show that consumer subsidy is more effective than manufacturer subsidy in promoting innovation investment. Sun et al. [49] study the issue of whether the government should subsidize the supply chain on the production side or the consumption side when the new energy vehicle industry is short of funds. The results show that, in order to maximize social welfare, the government has the highest subsidy when it adopts the consumption-side subsidy. For the subsidy polices in the GAPSC, Akkaya et al. [2] consider the fact that the producers have to go through a costly and low-yield transition period with uncertain future prospects when they experiment with and then adopt innovative production methods. On this basis, they scrutinize the effectiveness of different government policy instruments, i.e., taxes, subsidies, or both, in promoting agriculture innovation according to the real practices in Denmark. Alizamir et al. [50] compare the impacts of two government subsidy programs on consumers, farmers, and social warfare in the agriculture industry in the USA, i.e., the Price Loss Coverage program (price protection) and Agriculture Risk Coverage (revenue protection). However, all the above literature examines the roles of government subsidies in a static setting but not in a dynamic setting. 

Given that governments in various countries may adopt different subsidy policies to promote green innovation in the agriculture industry, without losing generality, this paper will still focus on manufacturer and consumer subsidies.

### 2.4. Research Gaps

Based on the provided literature review, we can find that the majority of the existing studies model GAPSC decision problems in static settings, except Luo et al. [23]. However, Luo et al. [23] do not take government subsidies into account. The studies with respect to government subsidies in green innovation under GAPSC also neglect the dynamic characteristics of supply chain operations. Furthermore, the important roles of the platform in GAPSC, i.e., traceability and DDM, deserve further investigation. The objective of this paper is to fill the above gaps. 

## 3. Problem Description and Model Hypothesis

Under the background of platform economy and green consumption, this paper considers a GAPSC composed of an agricultural product supplier (he) and an Internet service platform (she). The supplier produces two kinds of agricultural products simultaneously: green agricultural products and conventional agricultural products, which are sold at retail prices $p_{G}$ and $p_{C}$, respectively, with the help of the Internet platform. His decision-making task is to determine the level of green technology research and development (R&D) R(t), including the innovation of new green planting techniques such as improved irrigation methods and organic farming and the purchase of new equipment. Green agricultural products command a higher price in the market than conventional agricultural products due to the extra investment in green innovative technologies, i.e., $p_{G}>p_{C}$. It should be noted that both $p_{G}$ and $p_{C}$ are assumed to be constant in this paper. It is because the development of information technology and the enrichment of social networks provide consumers with various channels to obtain price information so that the price becomes more transparent, and then there is less space for price adjustment. In addition, each type of agricultural product (green or conventional) often exists in a completely competitive market. Thus, the suppliers can only passively accept the market prices.

The Internet platform has the functions of both product tractability and DDM. On one hand, she decides the green traceability level of the traceability system T(t), which may include the introduction cost, usage cost, and maintenance cost of two-dimensional code labels, QR codes, GPS scanning and positioning, and Internet of Things technologies such as radio frequency identification (RFID) and blockchain, so as to realize the whole process traceability and quality control for green agriculture products. On the other hand, the platform also needs to determine the level of DDM activities M(t) for both agricultural products, including digging deep into massive data, analyzing consumers’ purchasing preferences based on their historic consumption behaviors to identify the target consumers for featured sales, increasing the consumer conversion rate, and avoiding mistakes such as blind promotion and over-marketing.

The structure of the typical GAPSC with platform traceability and DDM, as shown in Figure 1.

**Assumption** **1.**
*In order to promote the development of GAPSC, the government generally chooses one of two typical green subsidy policies: consumer subsidy or supplier subsidy. In the first subsidy policy, the government subsidizes consumers who buy green agricultural products. Specifically, based on the unit sales price $p_{G}$ of green agricultural products, the government subsidizes $\left(1-\xi \right)p_{G}$ to consumers, and the unit price actually paid by consumers is $\xi p_{G}$, where ξ∈(0,1) denotes the proportion of unit sales price borne by consumers. This is an ordinary form of consumption subsidy in practice. In the second subsidy policy, the government subsidizes the supplier to enhance their motivation for carrying out green R&D. In particular, according to the supplier’s green investment cost $C_{R}$, the government provides the subsidies $\left(1-\phi \right)C_{R}$ for him; thus, the cost actually undertaken by the supplier is only $\phi C_{R}$, where ϕ is the proportion of green investment born by the supplier. This supplier subsidy form is also widely adopted in reality.*


**Assumption** **2.**
*The green degree is a comprehensive quantitative index to distinguish green agricultural products from conventional agricultural products. Nowadays, with the improvement of consumers’ requirements for the quality and safety of agricultural products and the enhancement of consumer environmental awareness, suppliers are committed to green R&D to improve the nutritional value and taste of agricultural products, reduce the content of harmful substances in agricultural products, and help reduce carbon emissions and promote environmental friendliness. Intuitively, suppliers’ green R&D efforts can directly improve the greenness of agricultural products. In addition, the Internet platform supervises and tracks the planting, production, and circulation of green agricultural products during the whole process through the Internet of Things and other technologies. This encourages the supplier to produce high-quality, high-security, and eco-friendly agricultural products and helps to eliminate adulteration in the circulation of agricultural products, providing a strong guarantee for dealing with the quality and environmental problems of green agricultural products and indirectly contributing to the greenness of agricultural products [51]. Referring to the modified goodwill model of EI Ouardighi [52] to characterize the joint influence of suppliers’ green R&D efforts and platform traceability input on the greenness of green agricultural products in the market, the differential equation of the greenness of green agricultural products can be expressed as follows:*

(1)
$$\left\{ \begin{array}{l} \dot{E}(t)=\mu_{S}R(t)+\mu_{P}T(t)-\sigma E(t)\\ E(0)=E_{0}\geq 0 \end{array}\right.$$

E(t)*represents the greenness of agricultural products at moment*t*, and*E(0)≥0*is the greenness of agricultural products at the initial time;*$\mu_{S}$*and*$\mu_{P}$*, respectively, represent influence coefficients of green R&D level*R(t)*and the traceability level*T(t)*on the greenness level of green agricultural products. What is more, with the rapid development of green technology, the standard of the greenness level of agricultural products is also constantly improving so that the agricultural products with high greenness in the past will gradually degenerate into low greenness even if other conditions remain unchanged. This can be regarded as the decline of agricultural products’ greenness with time, and it is assumed the greenness decays exponentially at the rate of*σ>0.

**Assumption** **3.**
*The supplier takes advantage of online marketing service provided by the Internet platform to sell agricultural products to consumers directly. In return, the supplier offers a certain percentage of commissions to the Internet platform based on sales of two kinds of agricultural products, namely $\varepsilon_{G}p_{G}D_{G}\left(t\right)$ and $\varepsilon_{C}p_{C}D_{C}\left(t\right)$, which is essentially a way of revenue sharing between the supplier and Internet platform. $\varepsilon_{G},\varepsilon_{C}>0$ are unit service commission rates charged by the Internet platform for green agricultural products and conventional agricultural products, respectively. For instance, JD.COM, T-mall and Amazon usually set a given service commission rate for each kind of product before sales, generally ranging from 0.5% to 10% of product sales. In this paper, we assume these two service commission rates are constant parameters.*


**Assumption** **4.**
*There are two distinct consumer segments in the market:*

λ

*denotes the proportion of consumers who prefer green agricultural products, and they believe that green agricultural products are superior to conventional agricultural products in health, taste, quality, and other aspects. The higher the greenness*

E(t)

*of agricultural products, the more favorable it is to stimulate the potential demand of this consumer segment. In contrast, the other*

(1−λ)

*proportion of consumer segments tend to conventional agricultural products, and the greenness level of agricultural products has no impact on the purchasing behaviors of this consumer segment. Moreover, the traceability investment of the Internet platform makes all links of production, processing, circulation and consumption of green agricultural products completely transparent, and consumers can get timely, accurate and complete information about them. This satisfies consumers’ requirement for traceability and enhances their trust in green agricultural products, which positively influences market demand for these products. Conducting DDM activities, realizing accurate marketing at the end of consumption and speeding up product circulation can enhance consumer demands for both agricultural products. Considering these two kinds of agricultural products are mutually substitutable, the linear demand model [53,54] is adopted, then the market demands for green agricultural products and conventional agricultural products are expressed as Equations (2) and (3), respectively.*

(2)
$$D_{G}(R(t),M(t),T(t))=\lambda \left[d-p_{G}+\gamma p_{C}+\beta_{1}M(t)+\eta T(t)+\theta E(t)\right]$$


(3)
$$D_{C}(M(t))=\left(1-\lambda \right)\left[d-p_{C}+\gamma p_{G}+\beta_{2}M(t)\right]$$


*In the above expressions, d is the market capacity and*

γ

*is the price sensitivity to the other kind of agriculture products, which reflects the substitutability between the two agricultural products.*

$\beta_{1}$

*and*

$\beta_{2}$

*are, respectively, the influence factors of DDM efforts*

M(t)

*on the demands for green agricultural products and conventional agricultural products;*

η

*and*

θ

*are, respectively, the influence coefficient of green traceability level of Internet platform*

T(t)

*and the greenness level*

E(t)

*on the demand for green agricultural products, which can embody consumers’ preference for green agricultural products.*


**Assumption** **5.**
*To avoid trivial cases, the fixed production cost of agricultural products is ignored here. Similar to [55], it is assumed that green technology does not affect the unit production cost of agricultural products. Additionally, following He et al. [56], the increasing quadratic functions are employed to measure the green R&D cost $C_{R}\left(t\right)$ for the supplier, green traceability investment $C_{T}\left(t\right)$ and DDM investment $C_{M}\left(t\right)$ for the platform, respectively, which can be expressed as follows:*

(4)
$$C_{R}\left(t\right)=\frac{k_{R}R^{2}\left(t\right)}{2},\;C_{T}\left(t\right)=\frac{k_{T}T^{2}\left(t\right)}{2},\;C_{M}\left(t\right)=\frac{k_{M}M^{2}\left(t\right)}{2}$$



$k_{R}$

*,*

$k_{T}$

*, and*

$k_{M}$

*, respectively, represent cost sensitivity coefficients of the green R&D level of suppliers, the green traceability level of the platform, and the DDM level of the platform.*


**Assumption** **6.**
*Since the supplier and the Internet platform respectively engage in green production and green traceability activities, i.e., they make green efforts together to ensure the greenness of agricultural products, we assume both two parties are of environmental consciousness. In other words, they not only care about their economic profits but also pay attention to environmental performance. The environmental performance in our model is reflected by the greenness level of green agricultural products. We introduce τ∈[0,1] to represent the relative importance degree of environmental concern utility to economic profit across the supply chain. In addition, in order to embody the differences between the two parties’ environmental consciousness degree, we also define χ∈[0,1] and 1−χ as the environmental concern coefficients of the platform and the supplier, respectively [56,57].*


**Assumption** **7.***The GAPSC can be operated for an infinite period. The supplier and the platform seek to maximize their economic profits and environmental utilities. The discount factors for both parties are*ρ>0.

Then, based on Assumptions 6 and 7, the objective functional of the supplier and the Internet platform can be expressed as follows:(5)$$U_{S}=\int_{0}^{\infty }e^{-\rho t}\left[\left[\left(p_{G}-\varepsilon_{G}p_{G}\right)D_{G}\left(t\right)+\left(p_{C}-\varepsilon_{C}p_{C}\right)D_{C}\left(t\right)-\frac{\phi k_{R}R^{2}\left(t\right)}{2}\right]+\left(1-\chi \right)\tau E\left(t\right)\right]\mathrm{dt}$$
(6)$$U_{P}=\int_{0}^{\infty }e^{-\rho t}\left[\left[\varepsilon_{C}p_{C}D_{C}\left(t\right)+\varepsilon_{G}p_{G}D_{G}\left(t\right)-\frac{k_{T}T^{2}\left(t\right)}{2}-\frac{k_{M}M^{2}\left(t\right)}{2}\right]+\chi \tau E\left(t\right)\right]\mathrm{dt}$$

The first brackets in formula $U_{S}$ and $U_{P}$, respectively, represent the profits of the supplier and the platform, and the second items (1−χ)τE(t) and χτE(t) are the environmental concern utilities of the supplier and the platform, respectively.

**Assumption** **8.**
*The decision-making process of the GAPSC can be regarded as a Stackelberg dynamic game. The supplier acts as the leader, and the platform serves as the follower.*


## 4. Model and Solutions

Based on the problem description and model assumptions in the previous section, this section models four scenarios of government subsidies for the GAPSC, namely non-government subsidy (NS), consumer subsidy (CS), supplier subsidy (SS), and supplier subsidy with green tractability cost sharing between the supplier and the platform (TSS), and then the optimal strategies are derived under each subsidy scenario. Hereinafter, the superscripts of variables and profit (utility) functions represent subsidy scenarios. 

### 4.1. Non-Government Subsidy (Model NS)

We first consider the no subsidy scenario as a benchmark. In this scenario, the supplier, as the leader, determines green R&D efforts by maximizing his utility first. The platform makes the optimal responses according to the strategy given by the supplier—that is, it determines the optimal green traceability and DDM levels to maximize her utility. The above decision-making process can be regarded as a Stackelberg dynamic game, and the differential game model is formulated as follows:(7)$$\begin{array}{l} \underset{R(\cdot )}{\max}\left\{J_{S}^{NS}=\int_{0}^{\infty }e^{-\rho t}\left[\left[\left(p_{G}-\varepsilon_{G}p_{G}\right)D_{G}\left(t\right)+\left(p_{C}-\varepsilon_{C}p_{C}\right)D_{C}\left(t\right)\right]+\left(1-\chi \right)\tau E\left(t\right)-\frac{k_{R}R^{2}\left(t\right)}{2}\right]\mathrm{dt}\right\}\\ \underset{T(\cdot ),M(\cdot )}{\max}\left\{J_{P}^{NS}=\int_{0}^{\infty }e^{-\rho t}\left[\left[\varepsilon_{C}p_{C}D_{C}\left(t\right)+\varepsilon_{G}p_{G}D_{G}\left(t\right)\right]+\chi \tau E\left(t\right)-\frac{k_{T}T^{2}\left(t\right)}{2}-\frac{k_{M}M^{2}\left(t\right)}{2}\right]\mathrm{dt}\right\}\\ \;s.t.\;\dot{E}(t)=\mu_{S}R(t)+\mu_{P}T(t)-\sigma E(t),E(0)=E_{0} \end{array}$$

**Proposition** **1.**
*In NS scenario,*
1.
*The optimal strategies for the supplier’s green R&D level, the platform’s green traceability, and DDM levels are as follows:*

$$R^{NS*}=\frac{\mu_{S}\left[\lambda \theta p_{G}\left(1-\varepsilon_{G}\right)+\left(1-\chi \right)\tau \right]}{k_{R}\left(\rho +\sigma \right)},T^{NS*}=\frac{\lambda \varepsilon_{G}p_{G}\eta \left(\rho +\sigma \right)+\mu_{P}\left(\lambda \varepsilon_{G}p_{G}\theta +\chi \tau \right)}{k_{T}\left(\rho +\sigma \right)},\;M^{NS*}=\frac{\left(1-\lambda \right)\varepsilon_{C}p_{C}\beta_{2}+\lambda \varepsilon_{G}p_{G}\beta_{1}}{k_{M}}$$

2.
*The temporal evolution rule of the greenness level of green agricultural products is:*


$E^{NS}\left(t\right)=\left(E_{0}-E_{\infty }^{NS}\right)e^{-\sigma t}+E_{\infty }^{NS}.$

*, where*

$$E_{\infty }^{NS}=\frac{\mu_{S}^{2}\lambda \left(p_{G}-\varepsilon_{G}p_{G}\right)\theta +\mu_{S}^{2}\left(1-\chi \right)\tau }{k_{R}\left(\rho +\sigma \right)\sigma }+\frac{\mu_{P}\lambda \eta \varepsilon_{G}p_{G}\left(\rho +\sigma \right)+\mu_{P}^{2}\left(\lambda \varepsilon_{G}p_{G}\theta +\chi \tau \right)}{k_{T}\left(\rho +\sigma \right)\sigma }$$

3.
*The utility of suppliers and platforms are:*

$$V_{S}^{NS}\left(t\right)=\frac{\lambda \theta \left(p_{G}-\varepsilon_{G}p_{G}\right)+\left(1-\chi \right)\tau }{\left(\rho +\sigma \right)}E\left(t\right)+\frac{1}{\rho }\left\{ \begin{array}{l} \frac{\left[\lambda \left(p_{G}-\varepsilon_{G}p_{G}\right)\beta_{1}+\left(1-\lambda \right)\left(p_{C}-\varepsilon_{C}p_{C}\right)\beta_{2}\right]\left[\left(1-\lambda \right)\varepsilon_{C}p_{C}\beta_{2}+\lambda \varepsilon_{G}p_{G}\beta_{1}\right]}{k_{M}}+\\ \frac{\left[\lambda \eta \left(p_{G}-\varepsilon_{G}p_{G}\right)\left(\rho +\sigma \right)+\mu_{P}\left[\lambda \theta \left(p_{G}-\varepsilon_{G}p_{G}\right)+\left(1-\chi \right)\tau \right]\right]\left[\lambda \varepsilon_{G}p_{G}\eta \left(\rho +\sigma \right)+\mu_{P}\left(\lambda \varepsilon_{G}p_{G}\theta +\chi \tau \right)\right]}{k_{T}{\left(\rho +\sigma \right)}^{2}}\\ +\frac{\mu_{S}^{2}{\left[\lambda \theta \left(p_{G}-\varepsilon_{G}p_{G}\right)+\left(1-\chi \right)\tau \right]}^{2}}{2k_{R}{\left(\rho +\sigma \right)}^{2}}+\left(1-\lambda \right)\left(p_{C}-\varepsilon_{C}p_{C}\right)\left(d-p_{C}+\gamma p_{G}\right)+\lambda \left(p_{G}-\varepsilon_{G}p_{G}\right)\left(d-p_{G}+\gamma p_{C}\right) \end{array}\right.$$


$$V_{P}^{NS}(t)=\frac{\lambda \varepsilon_{G}p_{G}\theta +\chi \tau }{\left(\rho +\sigma \right)}E(t)+\frac{1}{\rho }\left\{ \begin{array}{l} \frac{{\left[\left(1-\lambda \right)\varepsilon_{C}p_{C}\beta_{2}+\lambda \varepsilon_{G}p_{G}\beta_{1}\right]}^{2}}{2k_{M}}+\frac{{\left[\lambda \varepsilon_{G}p_{G}\eta \left(\rho +\sigma \right)+\mu_{P}\left(\lambda \varepsilon_{G}p_{G}\theta +\chi \tau \right)\right]}^{2}}{2k_{T}{\left(\rho +\sigma \right)}^{2}}+\\ \frac{\mu_{S}^{2}\left(\lambda \varepsilon_{G}p_{G}\theta +\chi \tau \right)\left[\lambda \theta \left(p_{G}-\varepsilon_{G}p_{G}\right)+\left(1-\chi \right)\tau \right]}{k_{R}{\left(\rho +\sigma \right)}^{2}}+\left(1-\lambda \right)\varepsilon_{C}p_{C}\left(d-p_{C}+\gamma p_{G}\right)+\lambda \varepsilon_{G}p_{G}\left(d-p_{G}+\gamma p_{C}\right) \end{array}\right.$$




**Proof.** See the Appendix A. □

### 4.2. Consumer Subsidy (Model CS)

As mentioned above, in the CS scenario, the government subsidizes $\left(1-\xi \right)p_{G}$ to consumers when they buy green agricultural products. Different from the NS scenario, the demands for green agricultural products and conventional agricultural products in this scenario are expressed as $D_{G}(t)=\lambda \left[d-\xi p_{G}+\gamma p_{C}+\beta_{1}M(t)+\eta T(t)+\theta E(t)\right]$ and $D_{C}(t)=\left(1-\lambda \right)\left[d-p_{C}+\gamma \xi p_{G}+\beta_{2}M(t)\right]$, respectively. The differential game is expressed as follows:(8)$$\begin{array}{l} \underset{R(\cdot )}{\max}\left\{J_{S}^{CS}=\int_{0}^{\infty }e^{-\rho t}\left[\left[\left(p_{G}-\varepsilon_{G}p_{G}\right)D_{G}(t)+\left(p_{C}-\varepsilon_{C}p_{C}\right)D_{C}(t)\right]+\left(1-\chi \right)\tau E(t)-\frac{k_{R}R^{2}(t)}{2}\right]\mathrm{dt}\right\}\\ \underset{T(\cdot ),M(\cdot )}{\max}\left\{J_{P}^{CS}=\int_{0}^{\infty }e^{-\rho t}\left[\left[\varepsilon_{C}p_{C}D_{C}(t)+\varepsilon_{G}p_{G}D_{G}(t)\right]+\chi \tau E(t)-\frac{k_{T}T^{2}(t)}{2}-\frac{k_{M}M^{2}(t)}{2}\right]\mathrm{dt}\right\}\\ \;s.t.\;\dot{E}(t)=\mu_{S}R(t)+\mu_{P}T(t)-\sigma E(t),E(0)=E_{0} \end{array}$$

**Proposition** **2.**
*In the CS scenario:*
1.*The optimal strategies for the supplier’s green R&D level, the platform’s green traceability, and DDM levels are as follows:*$R^{CS*}=\frac{\mu_{S}\left[\lambda \theta p_{G}\left(1-\varepsilon_{G}\right)+\left(1-\chi \right)\tau \right]}{k_{R}\left(\rho +\sigma \right)},$ $M^{CS*}=\frac{\left(1-\lambda \right)\varepsilon_{C}p_{C}\beta_{2}+\lambda \varepsilon_{G}p_{G}\beta_{1}}{k_{M}},$ $T^{CS*}=\frac{\lambda \varepsilon_{G}p_{G}\eta \left(\rho +\sigma \right)+\mu_{P}\left(\lambda \varepsilon_{G}p_{G}\theta +\chi \tau \right)}{k_{T}\left(\rho +\sigma \right)}$.2.*The dynamic evolution rule of the greenness level of green agricultural products is:* $E^{CS}\left(t\right)=\left(E_{0}-E_{\infty }^{CS}\right)e^{-\sigma t}+E_{\infty }^{CS}$, *where*$$E_{\infty }^{CS}=\frac{\mu_{S}^{2}\lambda \left(p_{G}-\varepsilon_{G}p_{G}\right)\theta +\mu_{S}^{2}\left(1-\chi \right)\tau }{k_{R}\left(\rho +\sigma \right)\sigma }+\frac{\mu_{P}\lambda \eta \varepsilon_{G}p_{G}\left(\rho +\sigma \right)+\mu_{P}^{2}\left(\lambda \varepsilon_{G}p_{G}\theta +\chi \tau \right)}{k_{T}\left(\rho +\sigma \right)\sigma }$$3.
*The utilities of the supplier and platform are:*

$$V_{S}^{CS}\left(t\right)=\frac{\lambda \left(p_{G}-\varepsilon_{G}p_{G}\right)\theta +\left(1-\chi \right)\tau }{\rho +\sigma }E\left(t\right)+\frac{1}{\rho }\left\{ \begin{array}{l} \frac{\left[\lambda \beta_{1}\left(p_{G}-\varepsilon_{G}p_{G}\right)+\left(1-\lambda \right)\beta_{2}\left(p_{C}-\varepsilon_{C}p_{C}\right)\right]\left[\lambda \beta_{1}\varepsilon_{G}p_{G}+\left(1-\lambda \right)\beta_{2}\varepsilon_{C}p_{C}\right]}{k_{M}}+\frac{\mu_{S}^{2}{\left[\lambda \left(p_{G}-\varepsilon_{G}p_{G}\right)\theta +\left(1-\chi \right)\tau \right]}^{2}}{2k_{R}{\left(\rho +\sigma \right)}^{2}}\\ \frac{\left[\lambda \eta \left(p_{G}-\varepsilon_{G}p_{G}\right)\left(\rho +\sigma \right)+\mu_{P}\left(\lambda \theta \left(p_{G}-\varepsilon_{G}p_{G}\right)+\left(1-\chi \right)\tau \right)\right]\left[\lambda \eta \varepsilon_{G}p_{G}\left(\rho +\sigma \right)+\mu_{P}\left(\lambda \varepsilon_{G}p_{G}\theta +\chi \tau \right)\right]}{k_{T}{\left(\rho +\sigma \right)}^{2}}+\\ \lambda \left(p_{G}-\varepsilon_{G}p_{G}\right)\left(d-\xi p_{G}+\gamma p_{C}\right)+\left(1-\lambda \right)\left(p_{C}-\varepsilon_{C}p_{C}\right)\left(d-p_{C}+\gamma \xi p_{G}\right) \end{array}\right.$$


$$V_{P}^{CS}\left(t\right)=\frac{\lambda \varepsilon_{G}p_{G}\theta +\chi \tau }{\rho +\sigma }E\left(t\right)+\frac{1}{\rho }\left\{ \begin{array}{l} \frac{{\left[\left(1-\lambda \right)\beta_{2}\varepsilon_{C}p_{C}+\beta_{1}\lambda \varepsilon_{G}p_{G}\right]}^{2}}{2k_{M}}+\frac{{\left[\eta \lambda \varepsilon_{G}p_{G}\left(\rho +\sigma \right)+\mu_{P}\left(\lambda \varepsilon_{G}p_{G}\theta +\chi \tau \right)\right]}^{2}}{2k_{T}{\left(\rho +\sigma \right)}^{2}}+\\ \frac{\mu_{S}^{2}\left(\lambda \varepsilon_{G}p_{G}\theta +\chi \tau \right)\left[\lambda \left(p_{G}-\varepsilon_{G}p_{G}\right)\theta +\left(1-\chi \right)\tau \right]}{k_{R}{\left(\rho +\sigma \right)}^{2}}+\left(1-\lambda \right)\varepsilon_{C}p_{C}\left(d-p_{C}+\gamma \xi p_{G}\right)+\lambda \varepsilon_{G}p_{G}\left(d-\xi p_{G}+\gamma p_{C}\right) \end{array}\right.$$




### 4.3. Supplier Subsidy (Model SS)

As mentioned above, in the SS scenario, the government provides the manufacturer with subsidy $\left(1-\phi \right)C_{R}$ to encourage his green R&D behavior. For this model, the differential game model is expressed as follows:(9)$$\begin{array}{l} \underset{R(\cdot )}{\max}\left\{J_{S}^{SS}=\int_{0}^{\infty }e^{-\rho t}\left[\left[\left(p_{G}-\varepsilon_{G}p_{G}\right)D_{G}(t)+\left(p_{C}-\varepsilon_{C}p_{C}\right)D_{C}(t)\right]+\left(1-\chi \right)\tau E(t)-\frac{\phi k_{R}R^{2}(t)}{2}\right]\mathrm{dt}\right\}\\ \underset{T(\cdot ),M(\cdot )}{\max}\left\{J_{P}^{SS}=\int_{0}^{\infty }e^{-\rho t}\left[\left[\varepsilon_{C}p_{C}D_{C}(t)+\varepsilon_{G}p_{G}D_{G}(t)\right]+\chi \tau E(t)-\frac{k_{T}T^{2}(t)}{2}-\frac{k_{M}M^{2}(t)}{2}\right]\mathrm{dt}\right\}\\ \;s.t.\;\dot{E}(t)=\mu_{S}R(t)+\mu_{P}T(t)-\sigma E(t),E(0)=E_{0} \end{array}$$

**Proposition** **3.**
*In the SS scenario:*
1.*The optimal strategies for the supplier’s green R&D level, the platform’s green traceability, and DDM levels are*:$R^{SS*}=\frac{\mu_{S}\left[\lambda p_{G}\left(1-\varepsilon_{G}\right)\theta +\left(1-\chi \right)\tau \right]}{\phi k_{R}\left(\rho +\sigma \right)}$, $T^{SS*}=\frac{\lambda \varepsilon_{G}p_{G}\eta \left(\rho +\sigma \right)+\mu_{P}\left(\lambda \varepsilon_{G}p_{G}\theta +\chi \tau \right)}{k_{T}\left(\rho +\sigma \right)}$ and $M^{SS*}=\frac{\lambda \varepsilon_{G}p_{G}\beta_{1}+\left(1-\lambda \right)\varepsilon_{C}p_{C}\beta_{2}}{k_{M}}$.2.*The dynamic evolution rule of the greenness level of green agricultural products is:* $E^{SS}\left(t\right)=\left(E_{0}-E_{\infty }^{SS}\right)e^{-\sigma t}+E_{\infty }^{SS}$, *where*$$E_{\infty }^{SS}=\frac{\mu_{S}^{2}\left[\lambda \left(p_{G}-\varepsilon_{G}p_{G}\right)\theta +\left(1-\chi \right)\tau \right]}{\phi k_{R}\left(\rho +\sigma \right)\sigma }+\frac{\mu_{P}\left[\lambda \varepsilon_{G}p_{G}\eta \left(\rho +\sigma \right)+\mu_{P}\left(\lambda \varepsilon_{G}p_{G}\theta +\chi \tau \right)\right]}{k_{T}\left(\rho +\sigma \right)\sigma }$$3.
*The utilities of the supplier and the platform are:*

$$V_{P}^{SS}(t)=\frac{\left(\lambda \varepsilon_{G}p_{G}\theta +\chi \tau \right)}{\rho +\sigma }E(t)+\frac{1}{\rho }\left\{ \begin{array}{l} \frac{{\left[\lambda \varepsilon_{G}p_{G}\beta_{1}+\left(1-\lambda \right)\varepsilon_{C}p_{C}\beta_{2}\right]}^{2}}{2k_{M}}+\frac{{\left[\lambda \varepsilon_{G}p_{G}\eta \left(\rho +\sigma \right)+\mu_{P}\left(\lambda \varepsilon_{G}p_{G}\theta +\chi \tau \right)\right]}^{2}}{2k_{T}{\left(\rho +\sigma \right)}^{2}}+\\ \frac{\mu_{S}^{2}\left(\lambda \varepsilon_{G}p_{G}\theta +\chi \tau \right)\left[\lambda \left(p_{G}-\varepsilon_{G}p_{G}\right)\theta +\left(1-\chi \right)\tau \right]}{\phi k_{R}{\left(\rho +\sigma \right)}^{2}}+\left(1-\lambda \right)\varepsilon_{C}p_{C}\left(d-p_{C}+\gamma p_{G}\right)+\lambda \varepsilon_{G}p_{G}\left(d-p_{G}+\gamma p_{C}\right) \end{array}\right.$$


$$V_{S}^{SS}(t)=\frac{\left[\lambda \left(p_{G}-\varepsilon_{G}p_{G}\right)\theta +\left(1-\chi \right)\tau \right]}{\left(\rho +\sigma \right)}E(t)+\frac{1}{\rho }\left\{ \begin{array}{l} \frac{\left[\lambda \left(p_{G}-\varepsilon_{G}p_{G}\right)\beta_{1}+\left(1-\lambda \right)\left(p_{C}-\varepsilon_{C}p_{C}\right)\beta_{2}\right]\left[\lambda \varepsilon_{G}p_{G}\beta_{1}+\left(1-\lambda \right)\varepsilon_{C}p_{C}\beta_{2}\right]}{k_{M}}+\frac{\mu_{S}^{2}{\left[\lambda \left(p_{G}-\varepsilon_{G}p_{G}\right)\theta +\left(1-\chi \right)\tau \right]}^{2}}{2\phi k_{R}{\left(\rho +\sigma \right)}^{2}}+\\ \frac{\left(\lambda \left(p_{G}-\varepsilon_{G}p_{G}\right)\eta \left(\rho +\sigma \right)+\mu_{P}\left[\lambda \left(p_{G}-\varepsilon_{G}p_{G}\right)\theta +\left(1-\chi \right)\tau \right]\right)\left[\lambda \varepsilon_{G}p_{G}\eta \left(\rho +\sigma \right)+\mu_{P}\left(\lambda \varepsilon_{G}p_{G}\theta +\chi \tau \right)\right]}{k_{T}{\left(\rho +\sigma \right)}^{2}}+\\ \left(1-\lambda \right)\left(p_{C}-\varepsilon_{C}p_{C}\right)\left(d-p_{C}+\gamma p_{G}\right)+\lambda \left(p_{G}-\varepsilon_{G}p_{G}\right)\left(d-p_{G}+\gamma p_{C}\right) \end{array}\right.$$




### 4.4. Supplier Subsidy with Green Traceability Cost Sharing (Model TSS)

As the platform does not directly contribute to the production of green agricultural products, the government rarely subsidizes it. However, the development of GAPSC requires not only the green R&D efforts of the supplier but also the platform’s green traceability efforts. To motivate the platform to engage in green traceability, larger suppliers often share a part of their green traceability investment with the platform, namely a cost-sharing contract [1]. Thus, we consider the scenario that the government subsidizes the supplier for its green production. Furthermore, the supplier and the platform make an agreement on a traceability cost-sharing contract. In this scenario, the supplier first decides the green R&D level and the optimal cost-sharing ratio ψ(t)∈(0,1), and then the platform determines the optimal green traceability and DDM levels. It also constitutes a Stackelberg differential game dominated by the supplier, and the objective function of both parties is formulated as follows: (10)$$\begin{array}{l} \underset{R(\cdot )}{\max}\left\{J_{S}^{TSS}=\int_{0}^{\infty }e^{-\rho t}\left[\left[\left(p_{G}-\varepsilon_{G}p_{G}\right)D_{G}(t)+\left(p_{C}-\varepsilon_{C}p_{C}\right)D_{C}(t)\right]+\left(1-\chi \right)\tau E(t)-\frac{\phi k_{R}R^{2}(t)}{2}-\frac{\psi (t)k_{T}T^{2}(t)}{2}\right]\mathrm{dt}\right\}\\ \underset{T(\cdot ),M(\cdot )}{\max}\left\{J_{P}^{TSS}=\int_{0}^{\infty }e^{-\rho t}\left[\left[\varepsilon_{C}p_{C}D_{C}(t)+\varepsilon_{G}p_{G}D_{G}(t)\right]+\chi \tau E(t)-\frac{\left(1-\psi (t)\right)k_{T}T^{2}(t)}{2}-\frac{k_{M}M^{2}(t)}{2}\right]\mathrm{dt}\right\}\\ \;s.t.\;\dot{E}(t)=\mu_{S}R(t)+\mu_{P}T(t)-\sigma E(t),E(0)=E_{0} \end{array}$$

**Proposition** **4.**
*In the TSS scenario,*
1.
*The optimal strategies for the supplier’s green R&D level, cost-sharing rate, and the platform’s green traceability and DDM levels are as follows:*

$$M^{TSS*}=\frac{\left(1-\lambda \right)\varepsilon_{C}p_{C}\beta_{2}+\lambda \varepsilon_{G}p_{G}\beta_{1}}{k_{M}},\;R^{TSS*}=\frac{\mu_{S}\left[\lambda \theta p_{G}\left(1-\varepsilon_{G}\right)+\left(1-\chi \right)\tau \right]}{\phi k_{R}\left(\rho +\sigma \right)}\;T^{TSS*}=\frac{\left[\left(2\lambda \eta p_{G}-\lambda \eta \varepsilon_{G}p_{G}\right)\left(\rho +\sigma \right)+\mu_{P}\left(2\lambda \theta p_{G}-\lambda \theta \varepsilon_{G}p_{G}+2\tau -\tau \chi \right)\right]}{2k_{T}\left(\rho +\sigma \right)}\;\psi =\frac{\left[\begin{array}{l} \left(2\lambda \eta p_{G}-3\lambda \eta \varepsilon_{G}p_{G}\right)\left(\rho +\sigma \right)+2\mu_{P}\left[\theta \lambda \left(p_{G}-\varepsilon_{G}p_{G}\right)+\left(1-\chi \right)\tau \right]\\ -\mu_{P}\left(\chi \tau +\theta \lambda \varepsilon_{G}p_{G}\right) \end{array}\right]}{\left[\begin{array}{l} \left(2\lambda \eta p_{G}-\lambda \eta \varepsilon_{G}p_{G}\right)\left(\rho +\sigma \right)+2\mu_{P}\left[\theta \lambda \left(p_{G}-\varepsilon_{G}p_{G}\right)+\left(1-\chi \right)\tau \right]\\ +\mu_{P}\left(\chi \tau +\theta \lambda \varepsilon_{G}p_{G}\right) \end{array}\right]}$$

*In order to ensure that the supplier’s cost-sharing ratio exists and is reasonable, certain conditions must be satisfied, as shown in*Table 1.2.*The dynamic evolution rule of the greenness level of green agricultural products is:* $E^{TSS}\left(t\right)=\left(E_{0}-E_{\infty }^{TSS}\right)e^{-\sigma t}+E_{\infty }^{TSS}$, where
$$E_{\infty }^{TSS}=\frac{\mu_{S}^{2}\left[\lambda \theta \left(p_{G}-\varepsilon_{G}p_{G}\right)+\left(1-\chi \right)\tau \right]}{\phi k_{R}\left(\rho +\sigma \right)\sigma }+\frac{\mu_{P}\left[\lambda \eta \left(2p_{G}-\varepsilon_{G}p_{G}\right)\left(\rho +\sigma \right)+\mu_{P}\left(\lambda \theta \left(2p_{G}-\varepsilon_{G}p_{G}\right)+\tau \left(2-\chi \right)\right)\right]}{2k_{T}\left(\rho +\sigma \right)\sigma }$$3.
*The utilities of the supplier and the platform are:*

$$\begin{array}{l} V_{S}^{TSS}\left(t\right)=\frac{\left[\theta \lambda \left(p_{G}-\varepsilon_{G}p_{G}\right)+\left(1-\chi \right)\tau \right]}{\left(\rho +\sigma \right)}E\left(t\right)\\ +\frac{1}{\rho }\left\{ \begin{array}{l} \frac{\left[\lambda \beta_{1}\left(p_{G}-\varepsilon_{G}p_{G}\right)+\left(1-\lambda \right)\beta_{2}\left(p_{C}-\varepsilon_{C}p_{C}\right)\right]\left[\varepsilon_{C}p_{C}\left(1-\lambda \right)\beta_{2}+\varepsilon_{G}p_{G}\lambda \beta_{1}\right]}{k_{M}}\\ +\frac{{\left[\lambda \eta \left(2p_{G}-\varepsilon_{G}p_{G}\right)\left(\rho +\sigma \right)+2\mu_{P}\left[\lambda \theta \left(p_{G}-\varepsilon_{G}p_{G}\right)+\left(1-\chi \right)\tau \right]+\mu_{P}\left(\chi \tau +\theta \lambda \varepsilon_{G}p_{G}\right)\right]}^{2}}{8k_{T}{\left(\rho +\sigma \right)}^{2}}\\ +\frac{\mu_{S}^{2}{\left[\lambda \theta \left(p_{G}-\varepsilon_{G}p_{G}\right)+\left(1-\chi \right)\tau \right]}^{2}}{2\phi k_{R}{\left(\rho +\sigma \right)}^{2}}+\left(d-p_{G}+\gamma p_{C}\right)\lambda \left(p_{G}-\varepsilon_{G}p_{G}\right)+\left(d-p_{C}+\gamma p_{G}\right)\left(p_{C}-\varepsilon_{C}p_{C}\right)\left(1-\lambda \right) \end{array}\right. \end{array}$$


(11)
$$\begin{array}{l} V_{P}^{TSS}\left(t\right)=\frac{\chi \tau +\theta \lambda \varepsilon_{G}p_{G}}{\left(\rho +\sigma \right)}E\left(t\right)\\ +\frac{1}{\rho }\left\{ \begin{array}{l} \frac{{\left[\left(1-\lambda \right)\varepsilon_{C}p_{C}\beta_{2}+\lambda \varepsilon_{G}p_{G}\beta_{1}\right]}^{2}}{2k_{M}}\\ +\frac{\left[\lambda \eta \varepsilon_{G}p_{G}\left(\rho +\sigma \right)+\mu_{P}\left(\chi \tau +\lambda \theta \varepsilon_{G}p_{G}\right)\right]\left[\lambda \eta p_{G}\left(2-\varepsilon_{G}\right)\left(\rho +\sigma \right)+2\mu_{P}\left[\lambda \theta \left(p_{G}-\varepsilon_{G}p_{G}\right)+\left(1-\chi \right)\tau \right]+\mu_{P}\left(\chi \tau +\theta \lambda \varepsilon_{G}p_{G}\right)\right]}{4k_{T}{\left(\rho +\sigma \right)}^{2}}\\ +\frac{\mu_{S}^{2}\left(\chi \tau +\theta \lambda \varepsilon_{G}p_{G}\right)\left[\lambda \theta \left(p_{G}-\varepsilon_{G}p_{G}\right)+\left(1-\chi \right)\tau \right]}{\phi k_{R}{\left(\rho +\sigma \right)}^{2}}+\varepsilon_{C}p_{C}\left(1-\lambda \right)\left(d-p_{C}+\gamma p_{G}\right)+\lambda \varepsilon_{G}p_{G}\left(d-p_{G}+\gamma p_{C}\right) \end{array}\right. \end{array}$$




**Proof.** See the Appendix A. □

## 5. Analysis of Model Results

### 5.1. Comparative Static Analysis

**Corollary** **1.**
*A comparative static analysis of the key parameters under the NS, CS, and SS subsidy scenarios is summarized in Table 2. See the Appendix A for specific analysis steps.*


It can be seen from Corollary 1 that: in three scenarios of NS, CS, and SS, (1) as the influence coefficient of greenness level on demand for green agricultural products θ and the proportion of green consumers λ, in order to meet consumers’ demand for green agricultural products, both the supplier and the platform will actively improve their green efforts, thus positively affecting the greenness level of green agricultural products and improving the environmental and social friendliness of the whole GAPSC. Therefore, the government and both parties should strive to improve consumers’ green awareness and increase the proportion of green consumers, which will enable them to have a healthier and lower-carbon living environment in the long run; (2) the enlargement of green consumers does not always lead to an increase in the platform’s DDM level. In fact, since the platform conduct DDM activities for both two types of agricultural products, whether the increase of λ could encourage the platform to improve the DDM level or not depends on the comparison of the platform’s marginal profits between the two types of products, i.e., if the green agricultural product can bring more benefit to the platform than conventional agricultural product ($\varepsilon_{G}p_{G}\beta_{1}>\varepsilon_{C}p_{C}\beta_{2}$), the platform will increase DDM level as λ increases, and vice versa; (3) the more sensitive consumers are to the green behavior of the platform, i.e., the greater of η, the more motivation the platform has to invest in green tractability. However, η has no influence on the supplier’s green production decision; (4) it can be seen that the relative importance of environmental concern τ will encourage both parties to increase their green investment, thus improving the greenness of green agricultural products. In addition, the higher the relative importance of environmental concern, the more their green efforts will be. However, the impact of the environmental concern coefficient of the platform χ on the greenness level is not monotonic, which depends on the comparisons between the two parties’ green investment efficiencies. Specifically, when the green investment efficiency of the platform $\mu_{P}^{2}/k_{T}$ is higher than that of the supplier $\mu_{S}^{2}/(\phi k_{R})$, the increase of χ has a positive impact on the greenness level of green agricultural products, and vice versa. It should be noted that the government subsidy can further improve the green investment efficiency of the supplier, i.e., the lower the proportion of the green R&D investment born by the supplier ϕ, the higher its green investment efficiency is; (5) the lower the cost-efficiency of green investment, i.e., the higher of $k_{R}$ or $k_{T}$ is, the lower both parties’ enthusiasms are for green investment, which will lead to the decrease of the greenness level of green agricultural products. However, the scaling parameter $k_{M}$ for DDM of the platform will not affect the greenness of green agricultural products; (6) the higher the price of green agricultural products, the more favorable it is for two parties to make green efforts and improve the greenness of green agricultural products. In contrast, the price of conventional agricultural products will only prompt the platform to increase its DDM efforts; and (7) the rapid development of Internet technology allows the platform to track and store customers’ purchase history information, such as location and preference, through information tracking tools and then implement a data-driven analysis. Thus, although the platform increasing the DDM efforts will not promote the green development of the agricultural products supply chain, it can help to improve the marketing efficiency of both agricultural products, and then contribute to the improvement of economic performance.

**Corollary** **2.***Under the TSS subsidy scenario, (a)* $\frac{\partial \psi }{\partial \tau }>0$; $\frac{\partial \psi }{\partial \chi }<0$; $\frac{\partial E^{TSS}}{\partial \chi }<0$; $\frac{\partial T^{TSS}}{\partial \chi }<0$; *(b) if*$\chi >\varepsilon_{G}$*, then*$\frac{\partial \psi }{\partial \lambda }>0$, $\frac{\partial \psi }{\partial \theta }>0$, $\frac{\partial \psi }{\partial \eta }>0$*; if*$\chi <\varepsilon_{G}$*, then*$\frac{\partial \psi }{\partial \lambda }<0$, $\frac{\partial \psi }{\partial \theta }<0$, $\frac{\partial \psi }{\partial \eta }<0$*; and (c) the impacts of the relevant parameters on*$R^{TSS}$*and*$T^{TSS}$*are the same as those under the NS, CS, and SS scenarios.*

**Proof.** See the Appendix A. □

It follows from Corollary 2 that: under the TSS scenario, the supplier will increase the green traceability cost-sharing ratio for the platform as the relative importance degree of environment concern utility to economic profit τ increases. However, different from the NS, CS, and SS scenarios, as the environmental concern coefficient of the platform χ increases, the supplier has less incentive to share the platform’s traceability cost, reducing the greenness of green agricultural products and the platform’s green traceability level. Thus, when the supplier provides a cost-sharing mechanism for the platform, the platform should not hold a relatively high environmental concern degree. 

Moreover, different from previous studies, we can also find that the optimal cost-sharing ratio ψ no longer exhibits a simple monotonous relation with the green consumers ratio λ, the demand sensitivity coefficient of greenness level θ**,** and the demand sensitivity coefficient of the platform’s traceability level. However, it depends on the relationship between the environmental concern coefficient of the platform χ and the unit service commission rate charged by the platform for green agricultural products $\varepsilon_{G}$. More specifically, when $\chi >\varepsilon_{G}$ ($1-\chi <1-\varepsilon_{G}$), that is, the platform’s environmental concern degree exceeds her service commission rate for green agricultural products, the increase of all of the influence factors of the green consumers ratio, and the demand sensitivity coefficients of the greenness and traceability levels can motivate the platform to improve the traceability level, and then induce the supplier to set a higher cost-sharing ratio. Otherwise, when $\chi <\varepsilon_{G}$ ($1-\chi >1-\varepsilon_{G}$), the increase in λ**,**
θ**,** and η causes the supplier to lower the cost-sharing ratio. 

The above findings are counterintuitive in that the supplier’s optimal cost-sharing ratio may not be consistent with his own environmental concern degree. The reason is as follows: when $\chi >\varepsilon_{G}$, i.e., $1-\chi <1-\varepsilon_{G}$, the platform’s marginal profit from selling green agricultural products is low, as is shown in Proposition 2. Even if the supplier’s environmental concern degree is lower than his earnings rate by selling green agricultural products through the platform’s agent selling, he has to enhance the cost-sharing ratio to guarantee that the platform’s green traceability level increases in λ, θ, and η; otherwise, the supplier’s own profit will be damaged due to the decrease in green traceability levels. Similarly, it is known that when $\chi >\varepsilon_{G}$, the supplier will maintain his own profitability by reducing the cost-sharing ratio in spite of the fact that his environmental concern degree is higher than his earnings rate of green agricultural products. Therefore, a higher cost-sharing ratio is driven by the platform’s green traceability efforts, whereas the supplier will choose the optimal cost-sharing ratio that maximizes his own utility, rather than taking the environmental benefit alone into consideration.

**Corollary** **3.**
*Under the CS scenario, the consumer subsidy has no impact on the green investment strategies of the supplier and the platform, as well as the greenness of agricultural products. However, it influences the utilities of both parties, as shown in Table 3.*


Corollary 3 suggests in the CS scenario that whether the consumer subsidy can increase the members’ profits or not depends on the competition intensity between two types of agricultural products γ. Specifically, if the price competition intensity is below a certain threshold, the consumer subsidy benefits both parties. On the contrary, if the price competition intensity is above the threshold, both parties suffer losses from the consumer subsidy. The threshold relies on the comparison between service commission rates for two types of agricultural products charged by the platform. If the unit service commission rate for conventional agricultural products $\varepsilon_{C}$ is lower (higher) than that for green agricultural products $\varepsilon_{G}$, the threshold is $\gamma_{1}=\frac{\lambda p_{G}\left(1-\varepsilon_{G}\right)}{\left(1-\lambda \right)\left(1-\varepsilon_{C}\right)p_{C}}$ ($\gamma_{2}=\frac{\lambda \varepsilon_{G}p_{G}}{\left(1-\lambda \right)\varepsilon_{C}p_{C}}$). In fact, in practice, such as Yimutian.com, the differences in unit service commission rates for the same category of agricultural products are very small or even negligible. Regardless of the relation between $\varepsilon_{C}$ and $\varepsilon_{G}$, when the price competition intensity γ is relatively low, the government should increase the consumer subsidy to reduce the price that the consumers need to pay for green agricultural products. An intuitive explanation of this finding is that the advantage of conventional agricultural products mainly lies in their low prices, and the high prices of green agricultural products tend to inhibit consumers’ purchasing intention. Thus, the green consumer subsidy can effectively mitigate the competition disadvantage of green agricultural products, expand their market demand, and then improve the profitability of the supplier and the platform. However, the consumer subsidy also has a negative impact on the demand for conventional agricultural products. Especially when the price competition is fierce, the consumer subsidy damages the normal competitive environment within the GAPSC and further intensifies the vicious competition between two types of agricultural products, and then inevitably hurts both parties’ interests. Therefore, the government should not adopt the CS strategy to avoid ineffective fiscal expenditure and prevent the profit losses of both parties when faced with a highly competitive agricultural product market. 

**Corollary** **4.***Under the SS scenario, the supplier’s green R&D level, the greenness level of green agricultural products, and the utilities of both parties all decrease with the increase of the proportion of the green investment born by the supplier* ϕ*, that is:*$$\frac{\partial R^{SS}}{\partial \phi }<0,\frac{\partial E_{\infty }^{SS}}{\partial \phi }<0,\frac{\partial V_{P\infty }^{SS}}{\partial \phi }<0,\frac{\partial V_{S\infty }^{SS}}{\partial \phi }<0$$

Corollary 4 provides the insight that under the SS scenario, the increasing government subsidy encourages the supplier to invest more in green R&D, which improves the greenness of green agricultural products and the utilities of both parties. Accordingly, the supplier subsidy policy contributes to inspiring a potential market for green agricultural products and promotes the development of GAPSC.

### 5.2. Comparative Analysis

In order to further analyze the influence of different subsidy strategies on the performance of GAPSC, this section compares four decision-making modes under a given government subsidy amount. The results and the corresponding management insights are as follows.

**Proposition** **5.***The steady-state values of the greenness level and optimal decisions under four subsidy scenarios satisfy:* $E_{\infty }^{NS*}=E_{\infty }^{CS*}<E_{\infty }^{SS*}<E_{\infty }^{TSS*}$; $R^{NS*}=R^{CS*}<R^{SS*}=R^{TSS*}$; $M^{NS*}=M^{CS*}=M^{SS*}=M^{TSS*}$; $T^{NS*}=T^{CS*}=T^{SS*}<T^{TSS*}$.

**Proof.** See the Appendix A. □

Proposition 5 clearly shows that compared to the NS scenario, the CS strategy cannot improve the supplier’s green R&D level and the greenness level of green agricultural products, while the SS strategy can. From this view, the SS strategy is more conducive to the green development of GAPSC. The above conclusion can easily be understood as follows: in the SS scenario, the supplier’s green behavior is directly stimulated by the government subsidy, while in the CS scenario, the government subsidy merely lowers the consumers’ purchasing expenses, but the supplier is not directly inspired. As a result, the greenness level of green agricultural products in the CS scenario is independent of the government subsidy. More importantly, compared to the SS strategy, the green traceability cost-sharing mechanism in the TSS strategy can further enhance the platform’s traceability level and the greenness level of green agricultural products. In contrast, the level of the platform’s DDM activities will not be affected by any subsidy strategy.

**Proposition** **6.**
*The demand of two types of agricultural products under different subsidy scenarios satisfy: (a) $D_{G\infty }^{NS}<D_{G\infty }^{CS}$, $D_{C\infty }^{NS}>D_{C\infty }^{CS}$, $\left\{ \begin{array}{l} D_{G\infty }^{NS}+D_{C\infty }^{NS}<D_{G\infty }^{CS}+D_{C\infty }^{CS},\;\mathrm{if}\;\gamma <\gamma_{3};\;\\ D_{G\infty }^{NS}+D_{C\infty }^{NS}\geq D_{G\infty }^{CS}+D_{C\infty }^{CS},\;\mathrm{otherwise}. \end{array}\right.$, where $\gamma_{3}=\frac{\lambda }{1-\lambda }$; (b) $D_{G\infty }^{NS}<D_{G\infty }^{SS}<D_{G\infty }^{TSS}$, $D_{C\infty }^{NS}=D_{C\infty }^{SS}=D_{C\infty }^{TSS}$, $D_{G\infty }^{NS}+D_{C\infty }^{NS}<D_{G\infty }^{SS}+D_{C\infty }^{SS}<D_{G\infty }^{TSS}+D_{C\infty }^{TSS}$.*


Proposition 6(a) indicates that compared to the NS scenario, the CS strategy boosts the demand for green agricultural products but decreases the demand for conventional agricultural products. If the price competition intensity between two types of agricultural products γ is less than a certain threshold $\gamma_{3}$, the increment of green agricultural product sales volume exceeds the decrement of conventional agricultural product sales volume, leading that the total demand in the CS scenario is higher than that of the NS scenario. Conversely, if γ is more than $\gamma_{3}$, the total demand of the CS scenario is lower than that of the NS scenario. In addition, the threshold $\gamma_{3}$ increases in the proportion of green consumers λ, i.e., the higher the ratio of green consumers to the market capacity, the more likely the CS strategy improves the total demand. 

Proposition 6(b) shows that both the SS and TSS strategies can improve the demand for green agricultural products and are free from reducing the demand for conventional agricultural products. Thus, these two strategies can improve the total demand for two types of products. Moreover, the TSS strategy can build on the SS strategy to further increase the total demand due to the fact that the cost-sharing mechanism enables the platform to invest more in green traceability and then attracts more consumers to purchase green agricultural products.

**Proof.** See the Appendix A. □

**Proposition** **7.**
*(a) When the price competition intensity between two types of agricultural products is lower than the threshold $\gamma_{1}$ (for $\varepsilon_{C}<\varepsilon_{G}$) or $\gamma_{2}$ (for $\varepsilon_{C}\geq \varepsilon_{G}$), the total system utilities under four different subsidy strategies satisfy: $\left\{ \begin{array}{l} V_{S\infty }^{NS}+V_{P\infty }^{NS}<V_{S\infty }^{CS}+V_{P\infty }^{CS}\\ V_{S\infty }^{NS}+V_{P\infty }^{NS}<V_{S\infty }^{SS}+V_{P\infty }^{SS}<V_{S\infty }^{TSS}+V_{P\infty }^{TSS} \end{array}\right.$; (b) When the price competition intensity between two types of agricultural products is higher than the threshold $\gamma_{1}$ (for $\varepsilon_{C}<\varepsilon_{G}$) or $\gamma_{2}$ (for $\varepsilon_{C}\geq \varepsilon_{G}$), the total system utilities under four subsidy strategies satisfy: $V_{S\infty }^{CS}+V_{P\infty }^{CS}<V_{S\infty }^{NS}+V_{P\infty }^{NS}<V_{S\infty }^{SS}+V_{P\infty }^{SS}<V_{S\infty }^{TSS}+V_{P\infty }^{TSS}$.*


**Proof.** See the Appendix A. □

From the perspective of the total system utility, the SS strategy always outperforms the NS strategy, while the CS strategy does not always benefit the GAPSC. Specifically, when the competition between two types of agricultural products is moderate, consumer subsidy is favorable to the improvement of system utility. On the contrary, consumer subsidy results in a loss in system utility when the competition between two types of products is fierce. They can be straightforward and are derived from Corollary 3 and Proposition 6. Thus, the government should be cautious about adopting the CS strategy according to the price competition intensity. Moreover, regardless of the parameter values, the system utility under the TSS scenario is strictly superior to that under the SS scenario, i.e., the traceability cost sharing mechanism between the supplier and the platform is always beneficial to the GAPSC. In conclusion, the SS strategy of the government is one of the critical steps in helping the supplier realize green transformation, up-gradation, and promoting the healthy development of GAPSC, while the cost sharing mechanism between the supplier and the platform can further improve the environmental and economic benefits of the GAPSC on the basis of the SS strategy. 

## 6. Numerical Examples

This section conducts numerical simulations and sensitivity analyses to verify the above conclusions and better understand the impacts of key system parameters on the decisions and profits in the GAPSC, and then illustrates the management insights. By referring to [56,58] and combined with the setting of this paper, we set the system parameters as follows: $p_{G}=1.6$, $p_{C}=0.6$, $k_{R}=0.4$, $k_{T}=0.4$, $k_{M}=0.4$, τ=0.5, θ=0.8, $\varepsilon_{G}=0.1$, σ=0.1, d=10, $\mu_{S}=0.6$, $\mu_{P}=0.5$, η=0.6, $\beta_{1}=0.5$, $\beta_{2}=0.5$, ρ=0.1, χ=0.5, γ=0.5, λ=0.2, and $\varepsilon_{G}=\varepsilon_{C}=0.1$. Considering that in the initial stage of green agriculture practice, the greenness of agricultural products is relatively low. Thus, without loss of generality, we assume $E_{0}=0$. It is easy to know this assumption does not affect the main observations of the models. In addition, in order to guarantee that the total government subsidy amounts under CS, SS and TSS scenarios are the same, we let ξ=ϕ=0.77.

Figure 2 demonstrates that the greenness level of green agricultural products and system utility under each scenario will converge to steady states over time. The greenness of green agricultural products and system utility under the TSS and NS strategies are the highest and the lowest, respectively. Furthermore, the SS strategy outweighs the CS strategy as the former significantly improves the greenness level of green agricultural products and the system utility, whereas the latter fails to enhance the greenness level and its system utility is just slightly greater than that of the NS strategy. The above phenomenon reveals that government intervention is a necessity in the early stages of green agriculture development because the GAPSC often faces a series of challenges, such as a lack of consumer green awareness and difficulties in green R&D. Under this situation, the best strategy is that, on the one hand, the government subsidizes the forerunner (the supplier) for his green R&D; on the other hand, the supplier should actively bear a portion of the platform’s green traceability cost.

Figure 3 shows the effects of price competition intensity and consumer subsidy ratio on demand for two types of agricultural products and two parties’ utilities under the CS scenario. Based on the expressions of two critical thresholds $\gamma_{3}$ and $\gamma_{2}$ in Propositions 6 and 7, we can calculate $\gamma_{3}=0.25$ and $\gamma_{2}=0.67$ through parameter values assignment. In Figure 3a, we can find that compared to the NS scenario, when the competition intensity between the two agricultural products is lower (higher) than 0.25, the incremental effect of the CS strategy on the demand for green agricultural products is sufficient (insufficient) to compensate for the loss of demand for conventional agricultural products due to the cannibalization effect, which increases the total demand in the market. Similarly, in Figure 3b,c, we know that compared to the NS scenario, when the competition intensity is lower (higher) than 0.67, the supplier and the platform will benefit (suffer a loss) from the consumer subsidy. Therefore, the government ought to attach great importance to the competition intensity before implementing the CS strategy. 

As can be observed in Figure 4, in the SS and TSS scenarios, the influence coefficients of the platform traceability level and greenness level on the demand for green agricultural products (η and $u_{P}$) positively affect the utilities of supply chain members. Particularly, in the early stages of green agriculture development, a relatively high price causes green agricultural products to be less popular than conventional agricultural products. However, with the increase of η and $u_{P}$, the platform’s traceability behavior contributes more significantly to the increase of a market demand for green agricultural products. Moreover, compared to the SS scenario, the greater η and $u_{P}$, the more remarkable the cost-sharing mechanism in the TSS strategy can improve the demand for green agricultural products and both parties’ utilities. 

Figure 5 examines the effects of the government subsidy on performances under different scenarios. We can first find in Figure 5a that with the increase of the government subsidy ratio, the CS strategy raises the demand for green agricultural products at the cost of the significant reduction of the demand for conventional agricultural products. In contrast, both the SS and TSS strategies are able to boost the demand for green agricultural products without decreasing the demand for conventional agricultural products. Figure 5b,c shows that compared to the NS scenario, the subsidy effects of all the CS, SS, and TSS strategies become more obvious as the subsidy ratios increase (note that γ=0.5 in Figure 5, and then the CS strategy is valid). Moreover, regardless of the subsidy ratio, the TSS strategy can achieve a win-win situation for both parties compared to the SS strategy due to the advantage of its cost-sharing mechanism. However, this advantage will gradually weaken as the subsidy ratio increases.

Figure 6 depicts the effects of the proportion of green consumers λ on the steady-state members’ utilities. It can be seen that under any subsidy strategy, both parities’ utilities increase with λ, which indicates that green agricultural products can bring more utilities to both parties than conventional agricultural products. In addition, the higher λ, the more significant benefits are caused by the cost-sharing mechanism under the TSS strategy.

Figure 7 shows the impacts of the influence coefficient of the greenness level on the demand for green agricultural products θ and the environmental concern coefficient of the platform χ on the steady-state utilities of both parties. The more sensitive consumers are to the greenness of green agricultural products, the higher utilities are gained by both parties. Therefore, the supplier and the platform should not only invest more in green R&D and green traceability to improve the greenness level by themselves, but also actively foster the consumers to develop green consumption awareness. For instance, the supplier can promote green agricultural products by means of advertising, media campaigns, and the platform’s DDM capabilities.

We can also find in Figure 7 that under all four scenarios, the increase of the environmental concern of the platform χ is always detrimental to the supplier. For the platform, if χ is below a certain threshold, her utility increases in χ; otherwise, the increase of χ instead leads to the decrease of her utility. Compared to other strategies, the increase of χ has a more significantly negative impact on both parities’ utilities under the TSS strategy. Particularly, when χ increases to a certain level, the utility of the platform under the TSS strategy is even lower than that of the NS scenario. The reason is that a relatively high environmental concern degree of the platform under the TSS strategy not only weakens the motivation of the supplier to make a green R&D investment, but also reduces his cost sharing ratio for green traceability. This causes a sharp decline in the greenness level and the demand for green agricultural products, and ultimately undermines both parties’ utilities. Therefore, we can easily deduce that the higher the environmental concern degree of the supplier, the more conducive to a smooth development of the GAPSC. 

## 7. Discussion and Conclusions

The final discussion and conclusions mainly present the contributions related to the results in this study. This study is significant for the sustainable development of GAPSC with platform traceability and DDM marketing under alternative government subsidy scenarios.

### 7.1. Discussion

The main objective of this paper is to explore the operational decisions of GAPSC under four government subsidy schemes of the NS, CS, SS, and TSS strategies. Additionally, this paper considers that the platform will not merely engage in simple selling activity, but will also invest in green traceability for green agricultural products and DDM activity for both green and conventional agricultural products.

Unlike the previous studies in GAPSC, where both parties only aimed at maximizing their immediate interests on the basis of the static game, this paper utilizes a differential game theory to establish Stackelberg differential game models under different subsidy scenarios, and then derives the optimal decisions with the purpose of their long-term utilities based on Bellman’s continuous dynamic programming method. Furthermore, we consider that the utilities of supply chain members are not only determined by their economic benefits but are the combinations of economic and environmental performances.

In addition, this paper investigates the conditions under which each subsidy strategy can achieve the improvements in economic and environmental performances of the GAPSC and then conducts comparisons between different subsidy strategies and reveals the impacts of key parameters. Different from previous studies which considered only SS or CS strategies from the government, we innovatively incorporate a cost-sharing contract mechanism between the supplier and the platform into the SS strategy. We find that this new TSS strategy is superior to both SS and CS strategies. Based on the analysis results in the above propositions and numerical examples, we propose the following policy recommendations for the government and both parties. For the government, it should give priority to subsidizing the supplier for his green R&D investment from the perspective of subsidy efficiency. Especially in the early stage of green agricultural development, supplier subsidy is a key link to guarantee the smooth and healthy development of GAPSC. If the supplier also shares a portion of the green traceability cost for the platform when he receives a subsidy from the government, the government is supposed to set a moderate subsidy rate, or a too-high subsidy rate will weaken the positive effect of cost-sharing mechanism. Moreover, the government should strategically adopt a CS strategy based on the competition intensity between two types of agricultural products. More specifically, it should not adopt a CS strategy to avoid ineffective fiscal expenditure and prevent the profit losses of system utility when faced with a highly competitive market. The supplier and the platform should actively strengthen the cultivation of the consumers’ green consumption awareness and improve consumers’ confidence in green agricultural products. In the long run, it will be conducive to the green development of GAPSC and enable consumers to have a greener and better living environment. For the supplier, it is profitable for him to provide financial support for the green traceability behavior of the platform on the basis of supplier subsidy and design an appropriate cost-sharing rate in TSS strategy according to market factors such as the platform’s service commission rate and the proportion of green consumers. More importantly, the supplier, as the leader of the GAPSC, is supposed to undertake more environmental responsibilities actively in each subsidy scenario. For the platform, she ought not to hold too-high environmental concern degree to prevent the failure of the TSS strategy. Furthermore, the platform should also determine DDM decisions in light of the marginal profits of two types of products. The above measures of both parties can not only reduce the expenditure of the government but also help to improve the economic and environmental performances of the GAPSC. In our opinion, the above policy recommendations also have certain applicability to the green production, green consumption, and sales strategy in other industrial green supply chains. 

### 7.2. Conclusions

This paper considers a two-level GAPSC consisting of one supplier and one Internet platform. Specifically, the supplier adopts a hybrid production mode, i.e., makes green R&D investments to produce green agricultural products besides conventional agricultural products. The platform implements green traceability for green agricultural products and DDM for both two types of products. From a dynamic perspective, the differential game models have established under four government subsidy scenarios: no subsidy (NS), consumer subsidy (CS), supplier subsidy (SS), and supplier subsidy with green traceability cost-sharing (TSS). Then the optimal feedback strategies under each scenario are derived based on Bellman’s continuous dynamic programming theory. After that, the impacts of key parameters on optimal decisions and both parties’ utilities are analyzed, and comparisons among different subsidy scenarios are conducted. Finally, several numerical examples are given to acquire more management insights. The main conclusions are as follows:

The optimal strategies under different scenarios depend on the relevant parameter values such as price competition intensity, the proportion of green consumers, both parties’ environmental concern coefficient and so on. In all four subsidy scenarios, the increase of the influence coefficient of greenness level on demand for green agricultural products and the proportion of green consumers are conducive to improving the greenness level and enhancing the environmental and economic performances of the GAPSC. However, the enlargement of green consumers does not always lead to the increase in the platform’s DDM level. Only when green agricultural products can bring more benefit to the platform than conventional agricultural products, the platform will improve the DDM level as the proportion of green consumers increases. 

Compared to the NS scenario, the CS strategy has twofold effects on the GAPSC: on the one hand, it encourages more consumers to buy green agricultural products; on the other hand, it reduces consumers’ willingness to purchase conventional agricultural products due to the substitutability between two types of products. In other words, the expansion of demand for green agricultural products comes at the expense of cannibalizing the market for conventional ones. Thus, the impacts of consumer subsidy on the total demand for agricultural products and both parties’ utilities rely on price competition intensity. When the price competition intensity is below a certain threshold, the consumer subsidy boosts the total market demand and benefits both two parties. On the contrary, if the price competition intensity is above the threshold, the market demand shrinks, and both parties suffer losses from the consumer subsidy. Moreover, the CS strategy cannot improve the supplier’s green R&D level and the greenness of green agricultural products.

Compared to the NS scenario, the SS strategy can always improve the supplier’s green R&D level, the greenness level and market demand for green agricultural products, as well as the system utility. The TSS strategy can build on the SS strategy to further make the supplier increase the green traceability cost-sharing ratio and the platform enhance the green traceability level due to the advantage of cost-sharing mechanism, which in turn yields a higher greenness level and more green agricultural products. Accordingly, a win-win situation for both parties can be realized. Moreover, the higher the proportion of green consumers or the influence coefficient of traceability level on the greenness level, the more significant benefits aroused from cost-sharing mechanism under the TSS strategy. In contrast, the positive effect of cost-sharing mechanism will be weakened as the subsidy ratio increases.

In all subsidy scenarios, the parameters regarding environmental concern are of vital importance to operational decisions and the subsidy effects. The relative importance of environmental concern will encourage both parties to improve their green investment, thus improving the greenness level of green agricultural products. However, the increased environmental concern coefficient of the platform is not always beneficial to improve the greenness level of the entire GAPSC. Under three scenarios of NS, CS, and SS, when the green investment efficiency of the platform is higher (lower) than that of the supplier, the increase of the environmental concern coefficient of the platform has a positive (negative) impact on the greenness level of green agricultural products. In contrast, under the TSS scenario, the increased environmental concern degree of the platform causes the supplier to have less incentive to share the platform’s traceability cost, thus reducing both parties’ intention in green investment and going against the improvement of greenness level. In aspects of both parties’ utilities, regardless of which subsidy scenario, the increase of the environmental concern degree of the platform is always detrimental to the supplier, while there exists an optimal environmental concern degree for the platform to maximize her utility. Compared to the other three strategies, the increase of the environmental concern degree of the platform is more unfavorable to both parties under the TSS strategy.

However, there are still some shortcomings in this paper. For example, this paper only considers a GAPSC with one supplier and platform. We can further examine the competition between two suppliers (e.g., one green agricultural products supplier and one conventional agricultural products supplier) on the operations decisions and performances in the GAPSC with platform traceability and DDM. In addition, the assumptions of this study are all based on information symmetry. However, in reality, the supplier or the platform tends to be more selfish and is reluctant to disclose their private information. Thus, future research may utilize asymmetric information game theory to explore the government’s subsidy strategy for a GAPSC with members’ private information.

## Figures and Tables

**Figure 1 ijerph-20-03056-f001:**
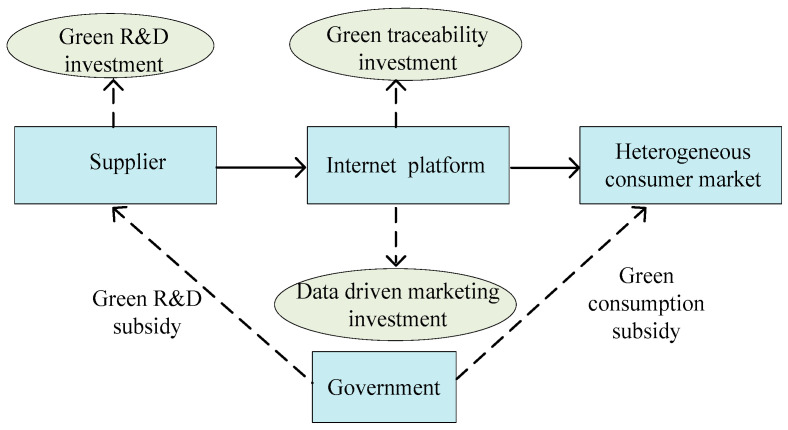
The structure of GAPSC with platform traceability and DDM.

**Figure 2 ijerph-20-03056-f002:**
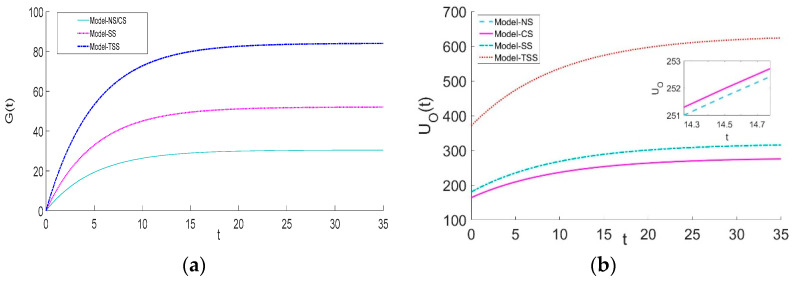
Time trajectory of the greenness level and system utility under four scenarios. (**a**) Greenness level. (**b**) System utility.

**Figure 3 ijerph-20-03056-f003:**
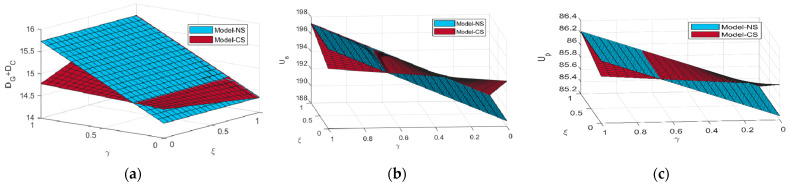
Effects of the competition intensity and the consumer subsidy ratio on performances under the CS scenario. (**a**) Total market demand for two types of agricultural products. (**b**) Supplier utility. (**c**) Platform utility.

**Figure 4 ijerph-20-03056-f004:**
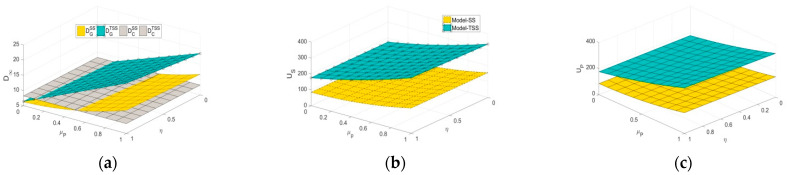
Impacts of green traceability factors η and $u_{P}$ on performances in the SS and TSS scenarios. (**a**) Market demand for two types of agricultural products. (**b**) Supplier utility. (**c**) Platform utility.

**Figure 5 ijerph-20-03056-f005:**
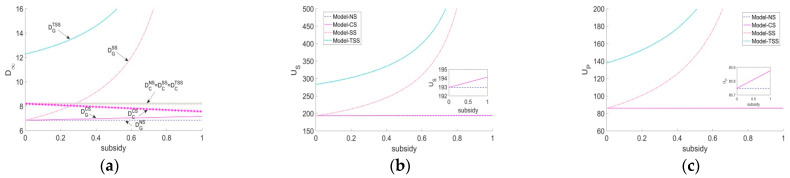
Effects of the government subsidy ratio on performances under different scenarios. (**a**) Market demand for two types of agricultural products. (**b**) Supplier utility. (**c**) Platform utility.

**Figure 6 ijerph-20-03056-f006:**
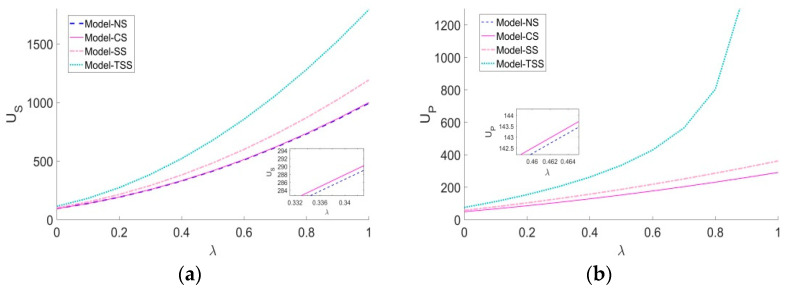
Effect of the proportion of green consumers λ on the utilities of supply chain members. (**a**) Supplier utility. (**b**) Platform utility.

**Figure 7 ijerph-20-03056-f007:**
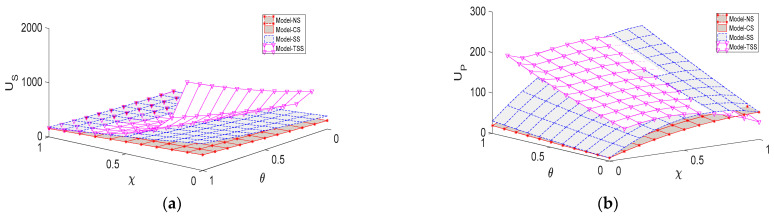
Impacts of the influence coefficient of the greenness level θ and the platform’s environmental concern coefficient χ on the utilities of supply chain members. (**a**) Supplier utility. (**b**) Platform utility.

**Table 1 ijerph-20-03056-t001:** Reasonable contract parameter region under the TSS model.

Unit Service Rate of the Platform	The Environmental Concern Coefficient of the Platform	The Relative Importance Degree of Environmental Concern utility to Economic Profit	Cost-Sharing Decision
$0<\varepsilon_{G}<\frac{2}{3}$	$0<\chi <\frac{2}{3}$	0<τ<1	Provide
	$\frac{2}{3}<\chi <1$	$\tau <\bar{\tau }$	Provide
	$\frac{2}{3}<\chi <1$	$\tau >\bar{\tau }$	Do not provide
$\frac{2}{3}<\varepsilon_{G}<1$	$0<\chi <\frac{2}{3}$	$\tau >\bar{\tau }$	Provide
	$0<\chi <\frac{2}{3}$	$\tau <\bar{\tau }$	Do not provide
	$\frac{2}{3}<\chi <1$	0<τ<1	Do not provide

where, $\bar{\tau }=\left[\lambda p_{G}\left[\eta \left(\rho +\sigma \right)+\mu_{P}\theta \right]\left(2-3\varepsilon_{G}\right)\right]/\left[\mu_{P}\left(3\chi -2\right)\right]$.

**Table 2 ijerph-20-03056-t002:** A comparative static analysis of the relevant parameters under the NS, CS, and SS subsidy scenarios.

	θ	λ	η	χ	τ	$\mu_{S}$	$\mu_{P}$	$k_{R}$	$k_{T}$	$k_{M}$	$p_{G}$	$p_{C}$
R	↗	↗	—	↘	↗	↗	—	↘	—	—	↗	—
T	↗	↗	↗	↗	↗	—	↗	—	↘	—	↗	—
M	—	↗,A>1;↘,A<1	—	—	—	—	—	—	—	↘	↗	↗
$E_{\infty }$	↗	↗	↗	$↗,\frac{\mu_{P}^{2}}{k_{T}}>\frac{\mu_{S}^{2}}{\phi k_{R}};↘,\frac{\mu_{P}^{2}}{k_{T}}<\frac{\mu_{S}^{2}}{\phi k_{R}}$	↗	↗	↗	—	—	—	↗	—

Note: $A=\frac{\varepsilon_{G}p_{G}\beta_{1}}{\varepsilon_{C}p_{C}\beta_{2}}$, ↗ indicates the positive influence, ↘ indicates the negative influence, and — indicates no influence.

**Table 3 ijerph-20-03056-t003:** Consumption subsidy conditions.

	$\varepsilon_{C}\geq \varepsilon_{G}$		$\varepsilon_{C}<\varepsilon_{G}$	
	$\gamma \geq \gamma_{2}$	$\gamma <\gamma_{2}$	$\gamma \geq \gamma_{1}$	$\gamma <\gamma_{1}$
$\partial \prod_{S}^{CS}/\partial \xi$	↗	↘	↗	↘
$\partial \prod_{P}^{CS}/\partial \xi$	↗	↘	↗	↘
Consumer subsidy	N	Y	N	Y

where $\gamma_{1}=\frac{\lambda p_{G}\left(1-\varepsilon_{G}\right)}{\left(1-\lambda \right)\left(1-\varepsilon_{C}\right)p_{C}}$ and $\gamma_{2}=\frac{\lambda \varepsilon_{G}p_{G}}{(1-\lambda )\;\varepsilon_{C}p_{C}}$; “Y” means “Yes” and “N” means “No”.

## Data Availability

Not applicable.

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
