# Peer review of "Green Agricultural Products Supply Chain Subsidy Scheme with Green Traceability and Data-Driven Marketing of the Platform"

_ijerph, 2023, doi:10.3390/ijerph20043056_

Round 1

Reviewer 1 Report

This is an excellent work that focuses on quantifying the different scenarios under which an agricultural supplier makes green research and development investments versus conventional processes, exploring platforms and various government subsidy scenarios. 

Authors have provided comprehensive details of all modelling components and good visualisations throughout. The appendix also provides further details, e.g. solving the Stackelberg model to derive the platforms green traceability levels. I think there are two things that could improve the paper further:

a) Around line 51, authors should stress the importance of data sharing in the context of agri-food, as that can be important in cases where information is needed to evaluate operations at various stages of the agri-food supply chain. A couple of references to add:

Durrant, A., Markovic, M., Matthews, D., May, D., Leontidis, G. and Enright, J., 2021. How might technology rise to the challenge of data sharing in agri-food?. Global Food Security28, p.100493.

Yan, B., Yan, C., Ke, C. and Tan, X., 2016. Information sharing in supply chain of agricultural products based on the Internet of Things. Industrial Management & Data Systems.

b) Conclusion is too long. Most of the stuff written in the conclusion should be part of a separate 'discussion' section and then leave the conclusion with a paragraph or two that summarises the main findings and provides the take away messages in a concise manner.

Author Response

Dear reviewer:

We are very grateful to your comments and suggestions for the manuscript entitled “Green Agricultural Products Supply Chain Subsidy Scheme with Green Traceability and Data-driven Marketing of the Platform” (ijerph-2188824). Those comments are all valuable and very helpful for revising and improving our paper, as well as the important guiding significance to our researches. We have studied comments carefully,summarized them and made correction which we hope meet with approval. Revised portion are marked in blue in the manuscript. The main corrections of the paper and responses to reviewer comments are as follows:

Point 1: Around line 51, authors should stress the importance of data sharing in the context of agri-food, as that can be important in cases where information is needed to evaluate operations at various stages of the agri-food supply chain. A couple of references to add:

Response 1: According to your suggestions, we have emphasized the importance of data sharing in the context of agri-food in the section describing platform traceability, see lines 71-75 for details.

Lines 71-75:The platform uses the Internet of Things, big data technology and visualization technology to share data and realize the traceability of agricultural products. It can not only scientifically and systematically manage the complex information of agricultural products, but also provide quality control of the whole process of agricultural products from farmland to table to ensure the safety of products [9-10].

Lines 1179-1182:

[9] Durrant, A.; Markovic, M.; Matthews, D.; May, D.; Leontidis, G.; Enright, J. How might technology rise to the challenge of data sharing in agri-food? Global Food Secur. 2021, 28, p.100493. https://doi.org/10.1016/j.gfs.2021.100493

[10] Yan, B., Yan, C., Ke, C. and Tan, X., Information sharing in supply chain of agricultural products based on the Internet of Things. Ind. Manage. Data Syst. 2016, 116(7) : 1397-1416.  https://doi.org/10.1108/IMDS-12-2015-0512

Point 2: Conclusion is too long. Most of the stuff written in the conclusion should be part of a separate 'discussion' section and then leave the conclusion with a paragraph or two that summarises the main findings and provides the take away messages in a concise manner.

Response 2: According to your suggestions, we have added the discussion part and streamlined and restructured the conclusion subsection in the original manuscript, which is specifically reflected in lines 844-893 and lines 918-958 of the paper.

Lines 844-893:The main objective of this paper is to explore the operational decisions of GAPSC under four government subsidy schemes of NS, CS, SS and TSS strategies. And this paper considers that the platform will not merely engage in simple selling activity, but will also invest in green traceability for green agricultural products and DDM activity for both green and conventional agricultural products.

Unlike the previous studies in GAPSC where both parties only aimed at maximizing their immediate interests on the basis of the static game, this paper utilizes differential game theory to establish Stackelberg differential game models under different subsidy scenarios, and then derives the optimal decisions with the purpose of their long-term utilities based on Bellman’s continuous dynamic programming method. What’s more, we consider that the utilities of supply chain members are not only determined by their economical benefits, but are the combinations of economic and environmental performances.

In addition, this paper investigates the conditions under which each subsidy strategy can achieve the improvements of economic and environmental performances of the GAPSC, and then conducts the comparisons between different subsidy strategies and reveals the impacts of key parameters. Different from previous studies which considered only SS or CS strategy from the government, we innovatively incorporate a cost-sharing contract mechanism between the supplier and the platform into SS strategy. We find that this new TSS strategy is superior to both SS and CS strategies. Based on the analysis results in the above propositions and numerical examples, we propose the following policy recommendations for the government and both parities. For the government, it should give priority to subsidizing the supplier for his green R&D investment from the perspective of subsidy efficiency.  Especially in the early stage of green agricultural development, supplier subsidy is a key link to guarantee the smooth and healthy development of GAPSC. If the supplier also shares a portion of green traceability cost for the platform when he receives subsidy from the government, the government is supposed to set a moderate subsidy rate, or a too high subsidy rate will weaken the positive effect of cost sharing mechanism. Moreover, the government should strategically adopt CS strategy based on the competition intensity be-tween two types of agricultural products. More specifically, it should not adopt CS strategy to avoid ineffective fiscal expenditure and prevent the profit losses of system utility when faced with a highly competitive market. The supplier and the platform should actively strengthen the cultivation of the consumers' green consumption awareness and improve consumers' confidence in green agricultural products. In the long run, it will be conducive to the green development of GAPSC and enable consumers to have a greener and better living environment. For the supplier, it is profitable for him to provide financial support for the green traceability behavior of the plat-form on the basis of supplier subsidy and design an appropriate cost sharing rate in TSS strategy according to market factors such as the platform’s service commission rate and the proportion of green consumers. More importantly, the supplier, as the leader of the GAPSC, is supposed to undertake more environmental responsibilities actively in each subsidy scenario. For the platform, she ought not to hold a too high environmental concern degree to prevent the failure of TSS strategy. Besides, the plat-form should also determine DDM decisions in the light of marginal profits of two types of products. The above measures of both parties can not only reduce the expenditure of the government, but also help to improve the economic and environmental performances of the GAPSC. In our opinion, the above policy recommendations al-so have a certain applicability to the green production, green consumption and sales strategy in other industrial green supply chains.

Lines 918-940: Compared to NS scenario, CS strategy has twofold effects on the GAPSC: on one hand, it encourages more consumers to buy green agricultural products; on the other hand, it reduces consumers' willingness to purchase conventional agricultural products due to the substitutability between two types of products. In other words, the expansion of demand for green agricultural products comes at the expense of cannibalizing the market for the conventional ones. Thus, the impacts of consumer subsidy on total demand of agricultural products and both parties’ utilities rely on price competition intensity. When the price competition intensity is below a certain threshold, the consumer subsidy boosts the total market demand and benefits both two parties. On the contrary, if the price competition intensity is above the threshold, the market demand shrinks and both parties suffer losses from the consumer subsidy. Moreover, CS strategy can not improve the supplier's green R&D level and the greenness of green agricultural products.

Compared to NS scenario, SS strategy can always improve the supplier's green R&D level, the greenness level and market demand for the green agricultural products, as well as the system utility. TSS strategy can build on SS strategy to further make the supplier increase the green traceability cost sharing ratio and the platform enhance the green traceability level due to the advantage of cost sharing mechanism, which in turn yields a higher greenness level and more green agricultural products. Accordingly, a win-win situation for both parties can be realized. Moreover, the higher the proportion of green consumers or the influence coefficient of traceability level on the green-ness level, the more significant benefits aroused from cost sharing mechanism under TSS strategy. By contrast, the positive effect of cost sharing mechanism will be weakened as the subsidy ratio increases.

 Finally, we appreciate for Reviewer’s enthusiastic work earnestly, and hope that the correction will meet with approval. Once again, thank you very much for your comments and suggestions.

Reviewer 2 Report

Dear authors, I think your piece is a very interesting economics exercise to show the benefits of greening agriculture. I think your exposition is rather complete and the literature review is well-developed. 

 I, however, think that you need to make a better job in presenting your results. I found the exposition is unclear and must be revised, in particular, to express the bnefits of your analysis.

I also think that the conclusion is too scarce. It is an economics application, which is fine but perhaps a critique and scope of your analysis will be welcomed by your readers.

Author Response

Dear reviewer:

We are very grateful to your comments and suggestions for the manuscript entitled “Green Agricultural Products Supply Chain Subsidy Scheme with Green Traceability and Data-driven Marketing of the Platform” (ijerph-2188824). Those comments are all valuable and very helpful for revising and improving our paper, as well as the important guiding significance to our researches. We have studied comments carefully,summarized them and made correction which we hope meet with approval. Revised portion are marked in blue in the manuscript. The main corrections of the paper and responses to reviewer comments are as follows:

Point: Authors need to make a better job in presenting your results. The exposition is unclear and must be revised, in particular, to express the benefits of your analysis. The conclusion is too scarce. It is an economics application, which is fine but perhaps a critique and scope of your analysis will be welcomed by your readers.

Response: According to your suggestions, we have revised the conclusion part, summarized our research results in more detail and added the discussion part. Specifically, we have revised lines 844-893 and 918-958 of the paper.

Lines 844-893:The main objective of this paper is to explore the operational decisions of GAPSC under four government subsidy schemes of NS, CS, SS and TSS strategies. And this paper considers that the platform will not merely engage in simple selling activity, but will also invest in green traceability for green agricultural products and DDM activity for both green and conventional agricultural products.

Unlike the previous studies in GAPSC where both parties only aimed at maximizing their immediate interests on the basis of the static game, this paper utilizes differential game theory to establish Stackelberg differential game models under different subsidy scenarios, and then derives the optimal decisions with the purpose of their long-term utilities based on Bellman’s continuous dynamic programming method. What’s more, we consider that the utilities of supply chain members are not only determined by their economical benefits, but are the combinations of economic and environmental performances.

In addition, this paper investigates the conditions under which each subsidy strategy can achieve the improvements of economic and environmental performances of the GAPSC, and then conducts the comparisons between different subsidy strategies and reveals the impacts of key parameters. Different from previous studies which considered only SS or CS strategy from the government, we innovatively incorporate a cost-sharing contract mechanism between the supplier and the platform into SS strategy. We find that this new TSS strategy is superior to both SS and CS strategies. Based on the analysis results in the above propositions and numerical examples, we propose the following policy recommendations for the government and both parities. For the government, it should give priority to subsidizing the supplier for his green R&D investment from the perspective of subsidy efficiency.  Especially in the early stage of green agricultural development, supplier subsidy is a key link to guarantee the smooth and healthy development of GAPSC. If the supplier also shares a portion of green traceability cost for the platform when he receives subsidy from the government, the government is supposed to set a moderate subsidy rate, or a too high subsidy rate will weaken the positive effect of cost sharing mechanism. Moreover, the government should strategically adopt CS strategy based on the competition intensity be-tween two types of agricultural products. More specifically, it should not adopt CS strategy to avoid ineffective fiscal expenditure and prevent the profit losses of system utility when faced with a highly competitive market. The supplier and the platform should actively strengthen the cultivation of the consumers' green consumption awareness and improve consumers' confidence in green agricultural products. In the long run, it will be conducive to the green development of GAPSC and enable consumers to have a greener and better living environment. For the supplier, it is profitable for him to provide financial support for the green traceability behavior of the plat-form on the basis of supplier subsidy and design an appropriate cost sharing rate in TSS strategy according to market factors such as the platform’s service commission rate and the proportion of green consumers. More importantly, the supplier, as the leader of the GAPSC, is supposed to undertake more environmental responsibilities actively in each subsidy scenario. For the platform, she ought not to hold a too high environmental concern degree to prevent the failure of TSS strategy. Besides, the plat-form should also determine DDM decisions in the light of marginal profits of two types of products. The above measures of both parties can not only reduce the expenditure of the government, but also help to improve the economic and environmental performances of the GAPSC. In our opinion, the above policy recommendations al-so have a certain applicability to the green production, green consumption and sales strategy in other industrial green supply chains.

Lines 918-958: Compared to NS scenario, CS strategy has twofold effects on the GAPSC: on one hand, it encourages more consumers to buy green agricultural products; on the other hand, it reduces consumers' willingness to purchase conventional agricultural products due to the substitutability between two types of products. In other words, the expansion of demand for green agricultural products comes at the expense of cannibalizing the market for the conventional ones. Thus, the impacts of consumer subsidy on total demand of agricultural products and both parties’ utilities rely on price competition intensity. When the price competition intensity is below a certain threshold, the consumer subsidy boosts the total market demand and benefits both two parties. On the contrary, if the price competition intensity is above the threshold, the market demand shrinks and both parties suffer losses from the consumer subsidy. Moreover, CS strategy can not improve the supplier's green R&D level and the greenness of green agricultural products.

Compared to NS scenario, SS strategy can always improve the supplier's green R&D level, the greenness level and market demand for the green agricultural products, as well as the system utility. TSS strategy can build on SS strategy to further make the supplier increase the green traceability cost sharing ratio and the platform enhance the green traceability level due to the advantage of cost sharing mechanism, which in turn yields a higher greenness level and more green agricultural products. Accordingly, a win-win situation for both parties can be realized. Moreover, the higher the proportion of green consumers or the influence coefficient of traceability level on the green-ness level, the more significant benefits aroused from cost sharing mechanism under TSS strategy. By contrast, the positive effect of cost sharing mechanism will be weakened as the subsidy ratio increases.

In all subsidy scenarios, the parameters regarding environmental concern is of vital importance to operational decisions and the subsidy effects. The relative im-portance of environmental concern will encourage both parties to improve their green investment, thus improving the greenness level of green agricultural products. How-ever, the increased environmental concern coefficient of the platform is not always beneficial to improve the greenness level of the entire GAPSC. Under three scenarios of NS, CS and SS, when the green investment efficiency of the platform is higher (low-er) than that of the supplier, the increase of the environmental concern coefficient of the platform has a positive (negative) impact on the greenness level of green agricultural products. By contrast, under TSS scenario, the increased environmental concern degree of the platform cause the supplier to have less incentive to share the platform’s traceability cost, thus reducing both parties’ intention in green investment and going against the improvement of greenness level. In aspects of both parties’ utilities, regard-less of which subsidy scenario, the increase of the environmental concern degree of the platform is always detrimental to the supplier; while there exists an optimal environmental concern degree for the platform to maximize her utility. Compared to other three strategies, the increase of the environmental concern degree of the platform is more unfavorable to both parties under TSS strategy.

Finally, we appreciate for Reviewer’s enthusiastic work earnestly, and hope that the correction will meet with approval. Once again, thank you very much for your comments and suggestions.

Reviewer 3 Report

Comments on IJERPH-2188824

This manuscript presents a potentially interesting and significant question, that is to investigate the most recent literature about green agricultural product supply chain (GAPSC) consisted of one supplier and one Internet platform. The supplier makes green R&D investment to produce green agricultural products along with conventional agricultural products, and the platform implements green traceability and data-driven marketing. The authors provided a reliable reference about the green agricultural products supply chain management, platform tractability and data-driven marketing. also, government subsidy for green innovation. The authors fail to present the importance of the study depending on available data. Discussion needs more and more amendments and, have to be more stand on the models four scenarios of government subsidies for the GAPSC. Also, the discussion needs more explanation’s for significant results. The conclusion section is confusing part. The readers cannot understand what exactly done with this study and what have been achieved from the results.

Line 31: Keywords: add new keywords and don’t repeated title words

L 1052: References:

Journal name (first letter must be Caps) e.g., lines, 1059, 1071

Journal number and page number must be added. e.g., lines, 1081, 1090, 1097, 1130, 1148, 1150, 1162.

Author Response

Dear reviewer:

We are very grateful to your comments and suggestions for the manuscript entitled “Green Agricultural Products Supply Chain Subsidy Scheme with Green Traceability and Data-driven Marketing of the Platform” (ijerph-2188824). Those comments are all valuable and very helpful for revising and improving our paper, as well as the important guiding significance to our researches. We have studied comments carefully,summarized them and made correction which we hope meet with approval. Revised portion are marked in blue in the manuscript. The main corrections of the paper and responses to reviewer comments are as follows:

Point 1: The authors fail to present the importance of the study depending on available data.

Response 1: According to your suggestions, we added existing data in the platform traceability and DDM parts to support our research. Specifically in lines 74-79, 83-87.

Lines 74-79:For example, "Ming Jing", a traceability platform for agricultural products, is based on cloud computing and cloud service technology, and uses the characteristics of block-chain, such as multi-participation and non-tampering, to build a data link that runs through all processes of agricultural production, processing and consumption. Through the combination of the Internet of Things and blockchain, it provides source reliable data support for breeding industry, production enterprises and consumers.

Lines 83-87:Yunnan Baiyao cooperates with Alibaba platform for data analysis, made use of big data and star effect for online marketing, and quickly improves its brand awareness. And Uniqlo realizes "zero inventory" through platform data analysis. Such marketing activities that utilize the platform data analysis capability are called platform data-driven marketing (DDM) activities.

Point 2: Discussion needs more and more amendments and, have to be more stand on the models four scenarios of government subsidies for the GAPSC.

Response 2: We added the discussion subsection in which we further highlight the contribution of this paper compared to previous studies, specifically in lines 844-893. In addition, management insights and policy recommendations are presented for members of GPASC in the context of four subsidy strategies. Specifically in lines 918-958.

Lines 844-893:The main objective of this paper is to explore the operational decisions of GAPSC under four government subsidy schemes of NS, CS, SS and TSS strategies. And this paper considers that the platform will not merely engage in simple selling activity, but will also invest in green traceability for green agricultural products and DDM activity for both green and conventional agricultural products.

Unlike the previous studies in GAPSC where both parties only aimed at maximizing their immediate interests on the basis of the static game, this paper utilizes differential game theory to establish Stackelberg differential game models under different subsidy scenarios, and then derives the optimal decisions with the purpose of their long-term utilities based on Bellman’s continuous dynamic programming method. What’s more, we consider that the utilities of supply chain members are not only determined by their economical benefits, but are the combinations of economic and environmental performances.

In addition, this paper investigates the conditions under which each subsidy strategy can achieve the improvements of economic and environmental performances of the GAPSC, and then conducts the comparisons between different subsidy strategies and reveals the impacts of key parameters. Different from previous studies which considered only SS or CS strategy from the government, we innovatively incorporate a cost-sharing contract mechanism between the supplier and the platform into SS strategy. We find that this new TSS strategy is superior to both SS and CS strategies. Based on the analysis results in the above propositions and numerical examples, we propose the following policy recommendations for the government and both parities. For the government, it should give priority to subsidizing the supplier for his green R&D investment from the perspective of subsidy efficiency.  Especially in the early stage of green agricultural development, supplier subsidy is a key link to guarantee the smooth and healthy development of GAPSC. If the supplier also shares a portion of green traceability cost for the platform when he receives subsidy from the government, the government is supposed to set a moderate subsidy rate, or a too high subsidy rate will weaken the positive effect of cost sharing mechanism. Moreover, the government should strategically adopt CS strategy based on the competition intensity be-tween two types of agricultural products. More specifically, it should not adopt CS strategy to avoid ineffective fiscal expenditure and prevent the profit losses of system utility when faced with a highly competitive market. The supplier and the platform should actively strengthen the cultivation of the consumers' green consumption awareness and improve consumers' confidence in green agricultural products. In the long run, it will be conducive to the green development of GAPSC and enable consumers to have a greener and better living environment. For the supplier, it is profitable for him to provide financial support for the green traceability behavior of the plat-form on the basis of supplier subsidy and design an appropriate cost sharing rate in TSS strategy according to market factors such as the platform’s service commission rate and the proportion of green consumers. More importantly, the supplier, as the leader of the GAPSC, is supposed to undertake more environmental responsibilities actively in each subsidy scenario. For the platform, she ought not to hold a too high environmental concern degree to prevent the failure of TSS strategy. Besides, the plat-form should also determine DDM decisions in the light of marginal profits of two types of products. The above measures of both parties can not only reduce the expenditure of the government, but also help to improve the economic and environmental performances of the GAPSC. In our opinion, the above policy recommendations al-so have a certain applicability to the green production, green consumption and sales strategy in other industrial green supply chains.

Lines 918-958: Compared to NS scenario, CS strategy has twofold effects on the GAPSC: on one hand, it encourages more consumers to buy green agricultural products; on the other hand, it reduces consumers' willingness to purchase conventional agricultural products due to the substitutability between two types of products. In other words, the expansion of demand for green agricultural products comes at the expense of cannibalizing the market for the conventional ones. Thus, the impacts of consumer subsidy on total demand of agricultural products and both parties’ utilities rely on price competition intensity. When the price competition intensity is below a certain threshold, the consumer subsidy boosts the total market demand and benefits both two parties. On the contrary, if the price competition intensity is above the threshold, the market demand shrinks and both parties suffer losses from the consumer subsidy. Moreover, CS strategy can not improve the supplier's green R&D level and the greenness of green agricultural products.

Compared to NS scenario, SS strategy can always improve the supplier's green R&D level, the greenness level and market demand for the green agricultural products, as well as the system utility. TSS strategy can build on SS strategy to further make the supplier increase the green traceability cost sharing ratio and the platform enhance the green traceability level due to the advantage of cost sharing mechanism, which in turn yields a higher greenness level and more green agricultural products. Accordingly, a win-win situation for both parties can be realized. Moreover, the higher the proportion of green consumers or the influence coefficient of traceability level on the green-ness level, the more significant benefits aroused from cost sharing mechanism under TSS strategy. By contrast, the positive effect of cost sharing mechanism will be weakened as the subsidy ratio increases.

In all subsidy scenarios, the parameters regarding environmental concern is of vital importance to operational decisions and the subsidy effects. The relative im-portance of environmental concern will encourage both parties to improve their green investment, thus improving the greenness level of green agricultural products. How-ever, the increased environmental concern coefficient of the platform is not always beneficial to improve the greenness level of the entire GAPSC. Under three scenarios of NS, CS and SS, when the green investment efficiency of the platform is higher (low-er) than that of the supplier, the increase of the environmental concern coefficient of the platform has a positive (negative) impact on the greenness level of green agricultural products. By contrast, under TSS scenario, the increased environmental concern degree of the platform cause the supplier to have less incentive to share the platform’s traceability cost, thus reducing both parties’ intention in green investment and going against the improvement of greenness level. In aspects of both parties’ utilities, regard-less of which subsidy scenario, the increase of the environmental concern degree of the platform is always detrimental to the supplier; while there exists an optimal environmental concern degree for the platform to maximize her utility. Compared to other three strategies, the increase of the environmental concern degree of the platform is more unfavorable to both parties under TSS strategy.

Point 3: The discussion needs more explanation’s for significant results.

Response 3: We have improved the explanation for the influences of ,  and on the optimal cost-sharing ratio in Corollary 2 and have given the reason why the advantages of TSS strategy over SS strategy shrinks as the government subsidy to supplier increases in Figure 5. Specifically in lines 628-641 and 805-808.

Lines 628-641: The above findings are counterintuitive in that the supplier’s optimal cost sharing ratio may not be consistent with his own environmental concern degree. The reason as follows:  when , i.e.,  the platform’s marginal profit from selling green agricultural products is low, as is shown in Proposition 2, even if the supplier’s environmental concern degree is lower than his earnings rate by selling green agricultural products through the platform’s agent selling, he has to enhance the cost-sharing ratio to guarantee that the platform's green traceability level increases in ,  and ; otherwise, the supplier’s own profit will be damaged due to the decrease of green traceability levels. Similarly, it is known that when  , the supplier will maintain his own profitability by reducing the cost-sharing ratio inspite of the fact that his environmental concern degree is higher than his earnings rate of green agricultural products. Therefore, the higher cost sharing ratio is driven by the platform’s green traceability efforts, whereas the supplier will choose the optimal cost sharing ratio that maximizes his own utility, rather than takes the environmental benefit alone into consideration.

Lines 805-808: Moreover, regardless of the subsidy ratio, TSS strategy can achieve a win-win situation for both parties compared to SS strategy due to the advantage of its cost sharing mechanism. However, this advantage will be gradually weakened as the subsidy ratio increases.

Point 4: The conclusion section is confusing part. The readers cannot understand what exactly done with this study and what have been achieved from the results.

Response 4: We have streamlined and restructured the conclusion subsection in the original manuscript. The specific modification is on lines 918-958.

Lines 918-958: Compared to NS scenario, CS strategy has twofold effects on the GAPSC: on one hand, it encourages more consumers to buy green agricultural products; on the other hand, it reduces consumers' willingness to purchase conventional agricultural products due to the substitutability between two types of products. In other words, the expansion of demand for green agricultural products comes at the expense of cannibalizing the market for the conventional ones. Thus, the impacts of consumer subsidy on total demand of agricultural products and both parties’ utilities rely on price competition intensity. When the price competition intensity is below a certain threshold, the consumer subsidy boosts the total market demand and benefits both two parties. On the contrary, if the price competition intensity is above the threshold, the market demand shrinks and both parties suffer losses from the consumer subsidy. Moreover, CS strategy can not improve the supplier's green R&D level and the greenness of green agricultural products.

Compared to NS scenario, SS strategy can always improve the supplier's green R&D level, the greenness level and market demand for the green agricultural products, as well as the system utility. TSS strategy can build on SS strategy to further make the supplier increase the green traceability cost sharing ratio and the platform enhance the green traceability level due to the advantage of cost sharing mechanism, which in turn yields a higher greenness level and more green agricultural products. Accordingly, a win-win situation for both parties can be realized. Moreover, the higher the proportion of green consumers or the influence coefficient of traceability level on the green-ness level, the more significant benefits aroused from cost sharing mechanism under TSS strategy. By contrast, the positive effect of cost sharing mechanism will be weakened as the subsidy ratio increases.

In all subsidy scenarios, the parameters regarding environmental concern is of vital importance to operational decisions and the subsidy effects. The relative im-portance of environmental concern will encourage both parties to improve their green investment, thus improving the greenness level of green agricultural products. How-ever, the increased environmental concern coefficient of the platform is not always beneficial to improve the greenness level of the entire GAPSC. Under three scenarios of NS, CS and SS, when the green investment efficiency of the platform is higher (low-er) than that of the supplier, the increase of the environmental concern coefficient of the platform has a positive (negative) impact on the greenness level of green agricultural products. By contrast, under TSS scenario, the increased environmental concern degree of the platform cause the supplier to have less incentive to share the platform’s traceability cost, thus reducing both parties’ intention in green investment and going against the improvement of greenness level. In aspects of both parties’ utilities, regard-less of which subsidy scenario, the increase of the environmental concern degree of the platform is always detrimental to the supplier; while there exists an optimal environmental concern degree for the platform to maximize her utility. Compared to other three strategies, the increase of the environmental concern degree of the platform is more unfavorable to both parties under TSS strategy.

Point 5: Line 31: Keywords: add new keywords and don’t repeated title words

Response 5: According to your comments, we have modified the keywords, specifically modified as follows.

Line 31:Keywords: Green agricultural products supply chain; Green subsidy; Platform traceability; Data-driven marketing; Differential game

Point 6: References:

Journal name (first letter must be Caps) e.g., lines, 1059, 1071

Journal number and page number must be added. e.g., lines, 1081, 1090, 1097, 1130, 1148, 1150, 1162.

Response 6: According to your comments, we have carefully checked all the references and capitalized the first letter of the journal name, such as references [4], [8], [11] and [29].

We have added the journal number and page number according to your requirements. For those references that have not found the journal number and page number, we added the doi link of the article.

[4] Yu, X.H.; Gao, Z.F.; Zeng, Y.C. Willingness to pay for the “Green Food” in China. Food Policy. 2014, 45, 80-87. https://doi.org/10.1016/j.foodpol.2014.01.003

[8] Akber, N.; Paltasingh, K.R.; Mishra, A.K. How can public policy encourage private investments in Indian agriculture? Input subsidies vs. public investment. Food Policy, 2022, 107, 102210. https://doi.org/10.1016/j.foodpol.2021.102210

[11] Aung, M.M.; Chang, Y.S. Traceability in a food supply chain: Safety and quality perspectives. Food Control. 2014, 39, 172-184. https://doi.org/10.1016/j.foodcont.2013.11.007

[29] Salah, K.; Nizamuddin, N.; Jayaraman, R.; Omar, M. Blockchain-based soybean traceability in agricultural supply chain. IEEE Access. 2019, 7, 73295-73305. https://doi.org/10.1109/ACCESS.2019.2918000

 Finally, we appreciate for Reviewer’s enthusiastic work earnestly, and hope that the correction will meet with approval. Once again, thank you very much for your comments and suggestions.

Reviewer 4 Report

It is an interesting study and very relevant. In different continents subsidies for the agricultural industry are large and may have a negative impact on the 'healthy' functioning of markets. The study gives a good overview of the potential approaches in subsidies; pro's and con's. Modeling the functioning of markets (and impacts) through gaming is a powerful contribution to scientific know-how and useful for government and other stakeholders in more sustainable agriculture.

Good conclusion: "TSS strategy can build on SS strategy to further enhance the 24 green traceability level of the platform, the greenness level and demand for green agricultural 25 products due to the advantage of cost-sharing mechanism. Accordingly, a win-win situation for both parties can be realized under TSS strategy".

Also a relevant conclusion is: "The optimal strategies under different scenarios depend on the relevant parameter values such as price competition intensity, the proportion of green consumers, both parties’ environmental concern coefficient and so on."

The paper could be improved by mentioning the limitations of the research approach when presenting conclusions.

Author Response

Dear reviewer:

We are very grateful to your comments and suggestions for the manuscript entitled “Green Agricultural Products Supply Chain Subsidy Scheme with Green Traceability and Data-driven Marketing of the Platform” (ijerph-2188824). Those comments are all valuable and very helpful for revising and improving our paper, as well as the important guiding significance to our researches. We have studied comments carefully,summarized them and made correction which we hope meet with approval. Revised portion are marked in blue in the manuscript. The main corrections of the paper and responses to reviewer comments are as follows:

Point: The paper could be improved by mentioning the limitations of the research approach when presenting conclusions.

Response: According to your comments, we have added the limitations of the research method in this paper in the last paragraph of the conclusion, which are as follows.

Lines 959-967: However, there are still some shortcomings in this paper. For example, this paper only considers a GAPSC with one supplier and one platform. We can further examine the competition between two suppliers (e.g., one green agricultural products supplier and one conventional agricultural products supplier) on the operations decisions and performances in the GAPSC with platform traceability and DDM. In addition, the assumptions of this study are all based on information symmetry. But in reality, the supplier or the platform tend to be more selfish and are reluctant to disclose their private information. Thus, future research may utilize asymmetric information game theory to explore the government's subsidy strategy for a GAPSC with members' private information.

 Finally, we appreciate for Reviewer’s enthusiastic work earnestly, and hope that the correction will meet with approval. Once again, thank you very much for your comments and suggestions.

Round 2

Reviewer 3 Report

No more comments